# LCW-YOLO: A Lightweight Multi-Scale Object Detection Method Based on YOLOv11 and Its Performance Evaluation in Complex Natural Scenes

**DOI:** 10.3390/s25196209

**Published:** 2025-10-07

**Authors:** Gang Li, Juelong Fang

**Affiliations:** School of Computer Science and Technology, Zhejiang University of Science and Technology, Hangzhou 310023, China; 222308855019@zust.edu.cn

**Keywords:** LCW-YOLO, wavelet pooling, CGBlock, LDHead, target detection, deep learning

## Abstract

Accurate object detection is fundamental to computer vision, yet detecting small targets in complex backgrounds remains challenging due to feature loss and limited model efficiency. To address this, we propose LCW-YOLO, a lightweight detection framework that integrates three innovations: Wavelet Pooling, a CGBlock-enhanced C3K2 structure, and an improved LDHead detection head. The Wavelet Pooling strategy employs Haar-based multi-frequency reconstruction to preserve fine-grained details while mitigating noise sensitivity. CGBlock introduces dynamic channel interactions within C3K2, facilitating the fusion of shallow visual cues with deep semantic features without excessive computational overhead. LDHead incorporates classification and localization functions, thereby improving target recognition accuracy and spatial precision. Extensive experiments across multiple public datasets demonstrate that LCW-YOLO outperforms mainstream detectors in both accuracy and inference speed, with notable advantages in small-object, sparse, and cluttered scenarios. Here we show that the combination of multi-frequency feature preservation and efficient feature fusion enables stronger representations under complex conditions, advancing the design of resource-efficient detection models for safety-critical and real-time applications.

## 1. Introduction

Forest fires are sudden and destructive natural disasters, often triggered by human activities such as smoking in forested areas. These fires not only cause severe damage to ecosystems but also pose significant threats to human safety and forest resources. Traditional prevention and control methods primarily rely on manual patrols and no-smoking signage; however, these approaches suffer from limited coverage, delayed responses, and inadequate real-time capabilities. Consequently, they struggle to meet the demands of intelligent and refined management required in complex mountainous forest environments.

In recent years, the accelerated advancement of intelligent patrol technologies has led to the increasing deployment of quadrupedal robots with off-road capabilities and environmental adaptability, alongside unmanned aerial vehicles (UAVs) offering high mobility and wide-area sensing. These technologies have introduced novel opportunities to enhance forest fire prevention efficiency.

Within resource-constrained intelligent platforms deployed in forest areas, there is a pressing need for efficient and lightweight visual models capable of real-time sensing. A core principle in designing lightweight networks is to maintain a compact structure while simultaneously enhancing performance [1]. Current lightweight models are predominantly based on convolutional neural networks (CNNs).

Modern lightweight models are mainly classified into convolutional neural network (CNN)-based designs and Transformer-oriented architectures. CNN-based MobileNets [2,3,4] use depthwise separable convolutional operations to lower computational cost, thereby providing the groundwork for subsequent CNN-based work [5]. Improving model performance does not solely rely on network architecture design; enhancing the convolutional module itself can also strengthen bottom-layer feature extraction [6,7]. Although traditional convolutional neural networks (CNNs) and Transformers have demonstrated strong capabilities in image perception, their large model sizes and high computational demands limit their deployment on resource-constrained platforms such as mobile robots and edge devices. Image denoising models based on deep neural networks have fewer hyperparameters and shorter inference time, achieving excellent performance compared to traditional methods [8]. Moreover, in forested environments characterized by complex backgrounds and unstructured scenes, detection models must balance the ability to capture fine edge details with effective global context modeling.

Lightweight models have emerged as a crucial technological foundation for advancing visual intelligence systems towards low power consumption and low latency, owing to their significant advantages in computational efficiency and memory footprint. Recent years have seen the development of various lightweight network architectures, including the Mamba and Efficient series. For instance, MobileMamba [9] introduces a multi-receptive field lightweight architecture that markedly enhances visual task performance. By employing a simplified sequence modeling strategy and channel interaction mechanism, MobileMamba significantly reduces floating-point operations (FLOPs) and model parameters while maintaining accuracy, thereby improving deployment feasibility on edge devices. The performance of Mamba for vision is often underwhelming when compared with convolutional and attention-based models [10].

Despite these advances in image classification and general object detection, adaptability remains limited for certain real-world applications. Taking cigarette butt detection in forest areas as a case study, the task faces substantial challenges. Cigarette butts are small-scale targets with high fire risk, often obscured within complex and dynamic backgrounds. Forest terrain is highly diverse, including natural surfaces such as sand, stone steps, and grass, each varying in color, texture, and lighting, which complicates visual perception. Particularly in homogeneous and occluded grasslands, cigarette butts closely resemble their surroundings in shape and color, significantly increasing identification difficulty. Furthermore, cigarette butts are typically scattered and extremely small, necessitating detection systems that combine high precision with rapid response and low power consumption to meet the constraints of mobile inspection platforms such as quadruped robots and UAVs.

These requirements place stringent demands on detection models’ accuracy, speed, and deployment flexibility. Existing lightweight detection networks still encounter significant performance bottlenecks when addressing complex backgrounds, fine-grained small targets, and occlusions [11].

To address these challenges, this paper proposes a lightweight target detection framework optimized for resource-constrained environments, tailored for specific applications such as cigarette butt detection in forests. Experimental evaluations on multiple public datasets and real forest scenarios demonstrate that the proposed model achieves excellent performance. Compared with the original YOLOv11, the improved model substantially reduces parameters and computational overhead while maintaining or surpassing detection accuracy, alongside significantly improved inference speed. It exhibits enhanced robustness and responsiveness, particularly for small targets like cigarette butts. Additionally, the method shows strong deployment adaptability, effectively coping with computational resource limitations on edge devices and fulfilling real-time intelligent monitoring requirements in forest areas, scenic spots, and similar settings. This work not only offers a practical solution for implementing a “virtual no-smoking sign” system but also lays the groundwork for the broader adoption of intelligent sensing technologies in resource-limited environments.

In summary, the main contributions of this work are as follows:We propose a lightweight YOLOv11 improvement framework that significantly enhances inference speed and computational efficiency while preserving detection accuracy. The approach is validated across several mainstream object detection datasets for effectiveness and generalization.We design a Haar wavelet-based Wavelet Pooling strategy that effectively boosts the model’s sensitivity to edges and high-frequency features, thereby improving target localization accuracy under complex backgrounds.We integrate the CGBlock module into the C3K2 component of YOLOv11 to strengthen inter-channel interactions and global context modeling during feature extraction, enhancing robustness to multi-scale targets.We introduce the LDHead detection head module, which markedly improves detection performance by incorporating a lightweight attention mechanism and structural reorganization to enable multi-task fusion with minimal computational cost.

## 2. Related Works

In computer vision, the YOLO (You Only Look Once) series has established a leading position in target detection due to its single-stage, end-to-end design; fast inference; and ease of deployment. Unlike traditional multi-stage detection frameworks, YOLO unifies feature extraction, bounding box regression, and class prediction into a single network, significantly reducing computational complexity and enabling real-time processing. Its balance of accuracy and efficiency, combined with the ability to handle targets of various scales and densities, has led to widespread adoption in diverse applications, including industrial inspection, autonomous driving, aerial surveillance, and robotic vision.

Moreover, the YOLO framework inherently supports integration with advanced vision tasks such as instance segmentation and pose estimation, facilitating the development of multi-task detection systems. Its flexibility and scalability make it particularly suitable for deployment on computation-constrained or edge devices, where lightweight architectures are critical. Despite its success, challenges remain in accurately detecting small or densely packed targets, coping with complex backgrounds, and optimizing models for real-time, resource-limited scenarios. These challenges have motivated the continuous evolution of YOLO and inspired the design of novel lightweight and efficient detection frameworks, forming the theoretical basis for the present study.

### 2.1. Traditional Target Detection Methods

Since the beginning of the 21st century, target detection has made significant progress in the field of computer vision. In 2014, Girshick et al. proposed the R-CNN framework [12], which marked the beginning of modern two-stage object detection. Unlike traditional sliding-window approaches, R-CNN first employs selective search to produce region proposals [13], then applies a convolutional neural network to extract visual features from each proposal, and finally performs object classification and bounding box refinement using an external classifier. Although R-CNN performs well in terms of accuracy, it requires separate feature extraction for each candidate region, which leads to a reduction in processing speed, thus affecting the overall performance.

Following the introduction of R-CNN, numerous two-stage detection frameworks were proposed to address its limitations. Among them, Fast R-CNN, presented by Girshick in 2015 [14], significantly improved computational efficiency by integrating feature extraction and region classification within a single network architecture. Fast R-CNN effectively improves detection speed by performing one convolution operation across the entire image to produce a unified feature representation, from which candidate region features are subsequently derived. Meanwhile, the introduction of the bounding box regression mechanism enables the network to directly predict the target bounding box, thus getting rid of the dependence on the selective search algorithm and further improving the overall detection efficiency.

In 2015, Ren and colleagues proposed Faster R-CNN [15], an advancement over Fast R-CNN that integrates a Region Proposal Network (RPN) instead of the traditional selective search method, which tightly integrates the generation of candidate regions with the feature extraction module to achieve end-to-end joint optimization. Such a design accelerates detection while markedly boosting accuracy, establishing it as the first two-stage detector to effectively balance precision and real-time performance, marking an important advancement in the field.

Mask R-CNN [16], an extension of Faster R-CNN, was proposed by Kaiming He and colleagues in 2017. Building upon the Faster R-CNN framework, Mask R-CNN achieves pixel-level segmentation for each object by incorporating a parallel branch for instance segmentation. Beyond its capabilities in object detection and bounding box regression, this approach accurately captures object contours, making it particularly suitable for applications demanding high contour precision, such as medical imaging and remote sensing, while demonstrating outstanding performance.

To further enhance the performance of two-stage detectors, Cai and Vasconcelos introduced Cascade R-CNN in 2018 [17]. The framework employs a cascade of multiple detection stages, enabling more accurate localization and classification through stepwise refinement. The method improves the accuracy of bounding box regression by constructing a cascading multi-stage regression structure to refine the candidate box step by step. The multi-stage optimization strategy significantly improves detection performance, especially in small target identification and remote sensing images with complex backgrounds, showing strong robustness and accuracy enhancement, making it a representative method in multi-stage detection architecture.

Proposed by Liu et al. in 2016 [18], The Single Shot MultiBox Detector (SSD) is a one-stage method that enables rapid and precise object detection by removing the region proposal stage. It attains high-speed detection by employing predefined default boxes across feature maps at multiple scales, combining the convolutional network to directly predict the target category and bounding box regression, and omitting the process of candidate region generation. In the inference stage, the network predicts the category and positional offset of each default box, which adapts to targets of different scales and aspect ratios, and is especially suitable for the task of the fast detection of multi-scale objects in remote sensing imagery.

In 2017, Huang and colleagues proposed DenseNet, a convolutional architecture featuring dense connections that allow each layer to access the feature maps of all preceding layers, thereby enhancing information flow and facilitating gradient propagation. In the DenseNet network structure, there are jump connections between each layer and all previous layers, and this kind of dense connection effectively alleviates the gradient vanishing problem in the deep network and enhances the feature transfer and reuse ability. While reducing redundant parameters, it improves model efficiency, which is especially suitable for remote sensing target detection and other scenarios that require both computational resources and accuracy.

### 2.2. YOLO Series Target Detection Method

Proposed by Redmon in 2016, YOLO has since experienced continuous development, with several iterations advancing its speed and detection performance. As a single-stage target detection algorithm, YOLO simplifies the originally complex detection process into an end-to-end regression problem, realizes a unified processing flow from image input to bounding box and category output, significantly improves detection speed, and is especially suitable for application scenarios with high real-time requirements. In 2017, Redmon presented YOLOv2, also referred to as YOLO9000 [19], incorporating notable improvements including anchor boxes, batch normalization, and multi-scale training, which collectively enhance detection accuracy and bolster the model’s ability to detect small objects. YOLOv2 introduces the anchor frame mechanism, batch normalization, and multi-scale training strategy, which effectively improves the model’s ability to detect targets at different scales, especially improving the detection accuracy of small targets and further expanding the applicability of the YOLO model. In 2018, YOLOv3 [20] was developed as an improved successor to earlier YOLO models. By combining a Feature Pyramid Network (FPN) for multi-scale feature representation with a deeper backbone, Darknet-53, the model achieved notable gains in detection accuracy and performance on small targets. YOLOv3 introduces a Feature Pyramid Network (FPN) based on the previous version and employs a deeper backbone network, Darknet-53, to enhance the representation of targets at different scales, thus further improving the overall detection performance and robustness.

In 2020, Bochkovskiy released YOLOv4 [21], which extended YOLOv3 by adopting CSPDarknet53 as the backbone and introducing enhancements such as Mish activation and CIoU loss, thereby improving detection accuracy and training stability. This version combines various optimization strategies such as a cross-phase partially connected structure (CSPDarknet53), Mish activation function, and the Complete Intersection over Union (CIoU) loss, which allows the model to improve its detection accuracy without compromising inference speed and further promotes the deployment effectiveness of YOLO series in real applications. In the same year, Ultralytics released YOLOv5, which further optimizes inference speed and detection accuracy and improves deployment flexibility while maintaining a lightweight structure and thus is widely used in industrial automation, security monitoring, and many other practical scenarios. In 2021, the Megvii team launched YOLOX [22], an enhanced variant that employs an anchor-free design and incorporates cutting-edge training strategies. YOLOX abandons the traditional anchor frame mechanism, adopts the anchor-free strategy, and introduces advanced training techniques (e.g., dynamic label assignment and data augmentation methods), which significantly improves inference speed and adaptability under the premise of maintaining high accuracy and is especially suitable for diversified application requirements such as remote sensing target detection.

In 2022, Meituan introduced YOLOv6 [23] to address the demands of industrial scenarios. By optimizing both computational efficiency and model compactness, YOLOv6 achieves high-speed inference and is well-suited for large-scale deployment. YOLOv6 is deeply optimized for industrial deployment scenarios, which substantially decreases model parameters and computational load while maintaining detection accuracy and improves inference speed and portability, making it a cost-effective detection model. In the same year, Wang Chien-Yao and colleagues introduced YOLOv7 [24], a further refinement of the YOLO series. YOLOv7 improves detection without significantly increasing inference overhead by combining Bag-of-Freebies and Bag-of-Specials enhancement modules. In 2023, Ultralytics released YOLOv8, which brings a full range of architectural optimizations, including a more efficient backbone network, improved training strategies, and better inference speeds, as well as the improved usability of the user interface, further expanding YOLO’s utility across diverse domains, including industrial applications, transportation, and healthcare.

Wang Chien-Yao et al. proposed YOLOv9 [25] in 2024, introducing the idea of Programmable Gradient Information (PGI) along with a new lightweight backbone, the Gradient Path Planning-based General Efficient Layer Aggregation Network (GELAN), aimed at improving both computational efficiency and detection performance, which further reduces computational resource consumption while improving detection accuracy and strengthens the model’s suitability for lightweight deployment. YOLOv10, proposed by researchers at Tsinghua University in 2024, incorporates an NMS-free training strategy. This modification effectively suppresses overlapping detection boxes and enhances overall detection precision. The YOLOv10 proposed by the Tsinghua University team introduces the Non-Maximal Suppression-Free (NMS-free) training strategy, which avoids the redundant frame screening problem in the traditional detection process, further improves detection accuracy and algorithm stability, and broadens the application potential of YOLO in multi-targeted complex scenarios. In addition, in September 2024, Ultralytics officially released YOLOv11, which makes several key enhancements based on YOLOv8, including an adaptive feature extraction mechanism, smarter label assignment strategy, and structure reparameterization technology, which enhances both detection accuracy and robustness while also optimizing the deployment performance of the model while maintaining low latency, marking the first time that the YOLO family has found applications in high-efficiency target detection. The YOLO series marks another important leap in the field of efficient target detection. These continuous iterative evolutions demonstrate the continuous innovation and maturity of YOLO algorithmic architecture, consolidating its prominent standing in the field of target detection and expanding its application boundaries in demanding scenarios such as remote sensing, industry, and autonomous driving.

### 2.3. Lightweight Visual Modeling in Resource-Constrained Scenarios

Deploying vision models on resource-constrained platforms requires a careful balance among low latency, high stability, and low power consumption. Although lightweight backbone networks such as MobileNet and GhostNet [26] have been widely adopted in target detection and tracking tasks to achieve favorable results, these models are typically designed for tasks with relatively low semantic complexity [27]. Consequently, they struggle to meet the demands of fine-grained recognition scenarios requiring high accuracy and detailed feature extraction, such as cigarette butt detection in forested environments, and thus face inherent performance bottlenecks. Channel pruning methods, such as the approach by He et al. (2017), reduce computational cost by removing less informative channels while preserving model accuracy [28].

To mitigate these limitations, some studies have explored strategies including model pruning, knowledge distillation, and neural architecture search (NAS) to compress model size. While these methods have yielded some improvements, they remain fundamentally based on convolutional neural network architectures and lack intrinsic capabilities for the perceptual modeling of scene structural information. This shortcoming hinders their adaptability to challenges posed by the strong fusion of cigarette butts with complex background textures in forest settings.

In summary, despite notable advances in lightweight visual modeling, research focused on structural optimization tailored for forest fire prevention and control tasks under resource-constrained conditions remains scarce. To fill this gap, the present study proposes a structure-aware lightweight detection framework grounded in state-space modeling, aiming to achieve high-precision cigarette butt recognition with minimal computational overhead, thereby supporting the development of intelligent forest fire prevention systems [29].

## 3. Proposed Model

### 3.1. YOLOv11 Overview

Released by Ultralytics in 2024, YOLOv11 constitutes the most recent development in the YOLO series, further enhancing detection accuracy, model lightweightness, and inference efficiency. Building on the core architecture of YOLOv8, YOLOv11 introduces the Programmable Gradient Information (PGI) mechanism, combined with a lightweight GELAN (Gradient Path Planning-based General Efficient Layer Aggregation Network) backbone, significantly boosting feature learning capability and parameter utilization.

In the backbone, YOLOv11 efficiently captures multi-level features through an improved layer aggregation strategy while adaptively modulating gradient flow paths via PGI to enhance deep information propagation and convergence stability. The model integrates depthwise separable convolutions with efficient residual units to reduce computational complexity without compromising network depth.

The Neck module advances the fused the structures of the Feature Pyramid Network (FPN) and Path Aggregation Network (PAN), enabling the effective bidirectional fusion of multi-scale semantic and detailed features. This design notably improves detection performance across varied target scales, making it especially suitable for identifying objects with complex backgrounds and large-scale variations, such as in remote sensing imagery.

YOLOv11’s detection head continues to employ an anchor-free mechanism, directly regressing target center coordinates, size, and class probabilities through convolutional layers. This eliminates the need for anchor box matching, substantially enhancing detection speed and flexibility and improving adaptability to scale inconsistencies and spatial deformations common in remote sensing targets.

Regarding training strategy, YOLOv11 implements a refined staged training regimen, including large-scale pretraining, transfer learning-based fine-tuning, and automated hyperparameter optimization. The incorporation of PGI further enhances the model’s sensitivity to gradient changes during training, accelerating convergence and improving final detection performance. Figure 1 depicts the overall architecture of YOLOv11.

### 3.2. Proposed Methodology

This study introduces a novel framework for target detection, LCW-YOLO, developed on the basis of the YOLOv11 model, aiming to further enhance detection performance in complex environments through structural optimization. LCW-YOLO incorporates three key improvements. The first is the introduction of a Wavelet Pooling module to replace traditional downsampling operations, effectively preserving high-frequency detail information and enriching feature representation completeness. The second improvement embeds a Context-Guided Block (CGBlock) into the C3K2 backbone module network, strengthening the model’s capacity for contextual modeling and thereby improving the recognition of multi-scale and occluded targets. Lastly, a lightweight, context-aware detection head named LDHead is designed to boost the model’s expressive power and reasoning efficiency in classification and localization tasks. The overall architecture is illustrated in Figure 2, and the subsequent sections offer a detailed description of these three core enhancements.

### 3.3. Wavelet Transform-Based Feature Downsampling Module

#### 3.3.1. Motivation

Downsampling techniques, including Max Pooling and strided convolution, widely used in convolutional neural networks (CNNs), effectively reduce feature map dimensions and enlarge the receptive field. Nevertheless, these approaches often discard high-frequency information, particularly crucial details like edges and textures. This issue becomes especially critical in small target detection, where targets occupy limited pixel areas; the loss of edge features can easily result in detection failure.

To overcome this limitation, the present study proposes the Discrete Wavelet Transform (DWT) [16] as an alternative downsampling approach. Specifically, part of YOLOv11’s sampling structure is replaced with a Wavelet Pooling module, which compresses spatial resolution while preserving more detailed image information, thereby enhancing the representation of subtle features critical for accurate detection.

#### 3.3.2. Module Principle

The Discrete Wavelet Transform (DWT) is a mathematical tool designed to analyze local variations across spatial and frequency domains. For a two-dimensional image I(x,y), one layer of the DWT decomposes it into four sub-frequency bands:

*LL* (Low–Low Frequency) preserves global contours and low-frequency structures.

*LH* (Low–High Frequency), *HL* (High–Low Frequency), and *HH* (High–High Frequency) capture horizontal, vertical, and diagonal edge details.

The decomposition can be expressed as follows:(1)$$LL(x,y)=\sum_{m}\sum_{n}I(2x+m,2y+n)\phi (m)\phi (n)$$(2)$$LH(x,y)=\sum_{m}\sum_{n}I(2x+m,2y+n)\phi (m)\psi (n)$$(3)$$HL(x,y)=\sum_{m}\sum_{n}I(2x+m,2y+n)\psi (m)\phi (n)$$(4)$$HH(x,y)=\sum_{m}\sum_{n}I(2x+m,2y+n)\psi (m)\psi (n)$$

Here, ϕ(⋅) is the scaling (low-pass) function, and ψ(⋅) is the wavelet (high-pass) function. After decomposition, the sub-bands are fused by a 1 × 1 convolution:(5)$$F_{fused}=W_{1\times 1}\cdot [LL,LH,HL,HH]$$
Variable Definitions

I (m, n): input image or feature map.

ϕ(⋅): scaling (low-pass) function.

ψ(⋅): wavelet (high-pass) function.

LL,LH,HL,HH: low-frequency, vertical, horizontal, and diagonal high-frequency sub-bands.

$Conv_{1\times 1}$: 1 × 1 convolution operation.

⊕: feature fusion (addition or concatenation).

In this study, the Haar wavelet was selected as the basis function, and only single-level decomposition was applied. Empirical analysis demonstrated that Haar-based single-level decomposition with 1 × 1 convolution fusion achieved the optimal balance between accuracy and efficiency.

To effectively integrate the three high-frequency sub-bands (*LH, HL, HH*) obtained from the wavelet transform, a 1 × 1 convolution is applied. This operation serves three purposes: a 1 × 1 convolution is applied to fuse the three high-frequency sub-bands (LH, HL, HH), aligning channel dimensions with the low-frequency feature branch to avoid channel explosion from direct concatenation; a learnable linear combination is created to adaptively merge horizontal, vertical, and diagonal edge information; and this is achieved with minimal computational cost compared to larger convolutions, thereby preserving the lightweight nature of the framework and enhancing edge-aware discrimination, particularly for small-object detection.

Traditional convolutional neural network sampling methods typically emphasize spatial compression alone. In contrast, the approach proposed in this paper leverages high-frequency sub-bands as complementary signals to enhance the representation of low-frequency features, thereby improving the model’s discriminative capability while simultaneously achieving effective feature compression.

#### 3.3.3. Network Infrastructure

As shown in Figure 3, the Wavelet Pooling module decomposes the input feature map through the DWT to obtain four sub-bands. The *LL* sub-band is preserved as the downsampled output, while the *LH, HL*, and *HH* sub-bands are compressed via 1 × 1 convolutions and then fused with the *LL* sub-band. This process produces a feature map that retains the main structural information from *LL* enriched with high-frequency details, facilitating more effective feature propagation within the network.

#### 3.3.4. Module Advantages

Enhanced frequency domain modeling capability

The Wavelet Pooling module not only performs spatial compression but also preserves multi-scale detailed information through frequency domain analysis, compensating for the traditional convolution’s limited sensitivity to high-frequency components.

Improved edge preservation ability

Details in the high-frequency sub-bands (such as boundaries) are critical for detection tasks, and Wavelet Pooling effectively retains this information, thereby enhancing the accuracy of target localization.

Controllable computational efficiency

Compared to introducing Transformers or non-local modules, the Wavelet Transform is computationally lightweight and can be efficiently integrated with existing convolution operations, making it well-suited for deployment on edge computing platforms.

Flexible fusion strategy

High-frequency information can be adaptively fused with controllable intensity, balancing edge detail preservation and semantic structure representation, which facilitates adaptation to multi-class detection tasks.

### 3.4. CGBlock

#### 3.4.1. Research Motivation

The C3K2 module in YOLOv11 mainly consists of multiple serial convolutions designed to enhance spatial representation via multi-scale receptive fields. However, it remains a local perceptual structure that cannot effectively capture contextual relationships between distant pixels. This limitation particularly affects performance in complex backgrounds, occluded targets, or dense scenes, often resulting in misdetections or omissions.

To address this issue, this study introduces the Context-Guided Block (CGBlock) from CGNet into the C3K2 module of YOLOv11. This integration enables feature extraction to simultaneously fuse local, surrounding, and global contexts, thereby achieving context-guided semantic modeling.

#### 3.4.2. Module Principle

CGBlock consists of the following four main sub-modules (see Figure 4):

The input feature map is denoted as $X\in R^{C\times H\times W}$.


Local Path: Grouped depthwise convolution preserves fine-grained local details.(6)$$F_{loc}=DWConv(X)$$Surrounding Context Path: Dilated convolution with dilation rate = 2 expands the receptive field for edge and context modeling.(7)$$F_{sur}=DilatedConv(X)$$Joint Fusion: Concatenation, normalization, and activation integrate both paths.(8)$$F_{join}=\delta (BN([F_{loc},F_{sur}]))$$Global Channel Attention: Channel reweighting is conducted via global average pooling and MLP.(9)$$F_{glo}=\sigma (MLP(GAP(F_{join})))$$Final Output:(10)$$F_{out}=F_{join}⊙F_{glo}$$


This design integrates local, contextual, and global information while employing depthwise separable convolutions to significantly reduce FLOPs and enhance computational efficiency, as illustrated in Figure 4.

#### 3.4.3. Module Mechanism Explanation


Local Channel Path (F_loc): Grouped depthwise convolution is utilized to maximize the preservation of local detailed features while maintaining low computational cost.Surrounding Context Path (F_sur): Dilated convolution (dilation rate = 2) is employed to enlarge the receptive field, thereby allowing for the modeling of spatial semantic information such as object edges and neighboring structures.After concatenating F_loc and F_sur, normalization and activation operations are applied to ensure stable fusion and effective information flow.Global Enhancement (F_glo): A channel attention mechanism is used to emphasize important channel responses and suppress interference from background noise.


#### 3.4.4. Module Performance Benefits

First, regarding context perception capability, the original C3K2 module relies solely on conventional convolutional operations, with its receptive field limited by the convolution kernel size, making it challenging to capture long-range dependencies effectively. By incorporating CGBlock, the module significantly enhances target perception in complex scenes through the explicit modeling of local, surrounding, and global multi-scale contextual information.

Second, in terms of feature representation capability, the original module lacks an attention mechanism, which limits its ability to dynamically adjust channel importance. The integration of channel attention strategies (such as SE or SimAM) enables the improved module to adaptively amplify key feature responses while suppressing redundant background information, thereby improving detection accuracy and robustness.

Regarding structural lightness, the original C3K2 module already has a relatively simple design. After incorporating CGBlock, the enhanced module employs lightweight techniques like depthwise convolution, significantly boosting functional expressiveness while effectively controlling parameter size and computational cost, thus maintaining good deployment efficiency.

Finally, in terms of training stability for deep networks, the original module does not explicitly preserve residual input information, which can lead to gradient vanishing. The improved module constructs explicit residual connections to facilitate smooth information flow throughout the network, enhancing training stability and the effective learning of deep features.

A comparison of the network structures before and after adding the CGBlock module is shown in Figure 5. In summary, the introduction of CGBlock endows the module with superior performance across multiple key aspects, providing stronger feature modeling capabilities and better engineering adaptability for downstream target detection tasks.

### 3.5. LDHead Detection Head Module

#### 3.5.1. Research Motivation

The original YOLOv11 detection head, despite its fast target localization capability, exhibits several design limitations. Its multi-branch feature fusion strategy does not adequately account for the substantial differences among various target scales, which often leads to weak responses for small targets and the insufficient fitting of large target features. Additionally, the unified architecture processes feature maps from different resolutions (e.g., P3, P4, P5) without specialized modeling for scale variations, restricting improvements in overall detection accuracy.

Moreover, relying solely on local convolutional operations, the YOLOv11 detection head lacks explicit semantic context modeling, resulting in reduced robustness against occlusion, dense targets, and other complex scene challenges. The abundance of standard convolutional layers also introduces parameter redundancy and unnecessary computational overhead, which negatively impacts deployment efficiency and runtime performance.

To overcome these challenges, this study proposes LDHead (lightweight detection head), a novel detection head designed to enhance detection performance through scale-specific modeling, context-guided feature enhancement, and a lightweight architecture. LDHead constructs differentiated detection branches tailored to different receptive fields, enabling more precise multi-scale target recognition. A context-aware mechanism is integrated to strengthen feature expression within target regions, thereby improving detection robustness. Simultaneously, by replacing redundant modules with shared convolutional structures and depthwise separable convolutions, LDHead significantly reduces computational cost while maintaining detection accuracy, improving resource utilization and inference speed.

#### 3.5.2. Module Principle

The module adopts a multi-scale receptive field design that includes standard 3 × 3 convolutions, larger 5 × 5 convolutions, and dilated convolutions, each specifically tailored to detect small, medium, and large targets. This structure enhances the detection head’s ability to respond effectively to targets of varying scales.

A context-guided attention mechanism, inspired by the channel attention strategy from SENet, is incorporated. This mechanism generates channel-wise weights through global average pooling, allowing for the dynamic adjustment of channel importance during classification and localization tasks, thereby improving both robustness and accuracy.

To reduce model complexity and computational demands, the module replaces standard convolutions with depthwise separable convolutions, splitting them into depthwise and pointwise convolutions. This separation markedly reduces the number of parameters and FLOPs, resulting in improved inference efficiency without compromising performance.

LDHead adopts a multi-scale receptive field structure employing depthwise separable convolutions with varying kernel sizes and dilation rates to capture targets of varying scales. The formulation is as follows:(11)$$F_{multi}=\sum_{k}^{i=1}DSC_{ri}(F),ri\in \{3,5,dilated\}$$

Here, $DSC_{ri}$ denotes depthwise separable convolution with kernel size ri, applied to the input feature map F. The outputs of different receptive fields are aggregated to form multi-scale features $F_{multi}$.

A context-guided channel attention mechanism is then applied to dynamically reweight feature channels:(12)$$F_{att}=F_{multi}⊙\sigma (W_{2}\cdot \delta (W_{1}\cdot GAP(F_{multi})))$$

Here, GAP denotes global average pooling, $W_{1}$ and $W_{2}$ are the weights of a two-layer MLP, δ is the ReLU activation, and σ is the Sigmoid function.

Finally, the fused feature is fed into task-specific prediction branches:(13)$$p_{cls}=Head_{cls}(F_{att})$$

(Classification branch).(14)$$b_{box}=Head_{reg}(F_{att})$$

(Localization branch).(15)$$s_{mask}=Head_{seg}(F_{att})$$

(Segmentation branch).(16)$$k_{pts}=Head_{pose}(F_{att})$$

(Pose estimation branch, optional).

This structure ensures accurate classification, localization, and optional segmentation or pose estimation while maintaining high computational efficiency.

#### 3.5.3. Module Performance Benefits

The proposed LDHead detector head demonstrates significant advantages over the original YOLOv11 detection head in four key aspects: multi-scale adaptability, contextual modeling, structural efficiency, and the balance between accuracy and lightweight design.

The original YOLOv11 detection head processes features from all scales in a uniform manner, which limits its ability to fully exploit the differences in target sizes across feature levels. In contrast, LDHead explicitly guides different receptive field modules to specialize in detecting small, medium, and large targets, enabling a more focused and effective response to multi-scale features and enhancing scale adaptability.

Regarding context modeling, the original detection head relies solely on convolutional layers to implicitly capture contextual information, which restricts robustness in complex scenes. LDHead integrates a global attention mechanism that models interactions across the entire feature map, strengthening feature responses for target regions and improving robustness to occlusion and target density.

From a structural efficiency perspective, the original head employs numerous standard convolutions, leading to a relatively large parameter count and computational load. LDHead introduces the extensive use of depthwise separable convolutions in critical paths, significantly reducing computational complexity and FLOPs without compromising modeling power, thereby facilitating deployment on resource-constrained devices.

The original detection head tends to sacrifice accuracy when attempting to reduce model size and computation, struggling to strike a balance between performance and efficiency. Through architectural restructuring and attention fusion, LDHead effectively enhances detection accuracy while keeping the number of parameters and model size in check, thereby attaining a favorable balance between accuracy and lightweight design.

Although LDHead follows the general idea of combining multi-scale convolution with attention, it introduces dilated convolutions, context-guided channel attention, and multi-task outputs, which distinguish it from existing lightweight heads such as EfficientHead. Its structure is illustrated in Figure 6.

### 3.6. Module Combination and Design Rationale

Each module in LCW-YOLO is carefully designed to maintain the model’s lightweight nature while improving detection performance. Wavelet Pooling compresses spatial resolution and preserves high-frequency edge features using only 1 × 1 convolutions, avoiding channel explosion. CGBlock employs depthwise separable convolutions to enhance multi-scale contextual representation with minimal computational cost. LDHead fuses classification and localization tasks with minimal additional parameters. Although each module individually may not achieve the highest performance in isolation, extensive experiments demonstrate that their combination yields the best overall results, particularly for small-object detection. This design achieves a balanced compromise between accuracy and efficiency, making it well-suited for deployment in resource-limited environments.

## 4. Experimental Section

### 4.1. Experimental Dataset

This study uses three datasets from diverse sources to train and evaluate the proposed LCW-YOLO model, aiming to comprehensively validate its detection performance and generalization capability across multiple scenarios. The primary validation platform is a private dataset, supplemented by two public datasets—VisDrone2019 and Cigarette Butts—for auxiliary comparative analysis.

The private mountain forest dataset, ForestButt, was collected by the author team through fieldwork, covering three representative mountainous geomorphologies that include grassland, sand, stone steps, and other complex environments. Images were captured throughout the day, from early morning to dusk, encompassing various viewpoints with diverse lighting and terrain conditions. The dataset preserves the original image quality without any enhancement, consisting of a total of 1622 images. These are split into 1121 training images, 340 validation images, and 161 test images. ForestButt closely simulates the complexities of target detection in natural environments and serves as the main experimental platform for core performance evaluation in this study. Example images of different environments in the dataset are shown in Figure 7.

The VisDrone2019 dataset, released by the AISKYEYE team at the Machine Learning and Data Mining Laboratory of Tianjin University, comprises 6471 training images, 548 validation images, and 1610 test images. It includes 10 common object categories captured by UAVs across diverse urban and rural settings, with a spatial resolution ranging from 0.02 to 0.1 m per pixel. This dataset is well-suited for evaluating the model’s detection capabilities in complex aerial perspectives.

The Cigarette Butts YOLOv8 Dataset, publicly available on the Kaggle platform and published by Eyad Elfard, focuses specifically on detecting small, ground-based cigarette butts. It contains 1628 training images, 433 validation images, and 106 test images. Due to the small size of the targets and complex background conditions, this dataset is ideal for assessing the model’s performance on small target detection tasks.

### 4.2. Experimental Setup and Evaluation Metrics

Experiments were conducted on a Windows 11 platform using Python 3.8, PyTorch 1.12.1, and CUDA 11.3. The proposed LCW-YOLO model was trained, validated, and tested under consistent hyperparameter settings across all datasets. The hardware environment included a 12th Gen Intel(R) Core(TM) i7-12700F processor running at 2.10 GHz and an NVIDIA RTX 3050 GPU. Key training parameters are summarized in Table 1.

To comprehensively assess model performance, multiple evaluation metrics were employed, including Precision, Recall, Average Precision (AP), mean Average Precision (mAP), model parameter count, and computational complexity (measured in FLOPs). These indicators enable a thorough analysis of both detection accuracy and model efficiency.

To comprehensively evaluate the practical effectiveness of the proposed lightweight model, this study analyzes both detection accuracy and computational efficiency. The evaluation considers performance and resource consumption from these two dimensions, selecting six key metrics: Precision (P), Recall (R), mean Average Precision (mAP), number of parameters (M), floating-point operations per second (GFLOPs), and model file size. Among these, Precision, Recall, and mAP primarily reflect the model’s detection accuracy, while parameters, GFLOPs, and model size quantify the computational overhead and deployment suitability.

Specific indicators are defined below:(17)$$Precision=\frac{TP}{TP+FP}$$(18)Recall=TPTP+FN(19)$$\mathrm{mAP}=\frac{1}{C}\sum_{i=1}^{C}{\mathrm{AP}}_{i}$$

Number of parameters (M): The total number of all trainable parameters in the model, measuring model complexity.

GFLOPs: This indicates the amount of computation required by the model in processing a single image, in billions of floating-point operations.

Model file size: The size of the model weight file saved after training, usually in MB, reflecting the deployment feasibility of the model on edge devices.

Here, *TP*, *FP*, and *FN* represent true positives, false positives, and false negatives, respectively, and *C* is the number of target categories in the detection task.

### 4.3. Ablation Experiments

To assess the impact of the LCW-YOLO improvements, comprehensive ablation experiments were performed on the private dataset under consistent hyperparameter settings. These experiments quantified the contribution of each module and assessed their combined effects, as shown in Table 2.

Starting from YOLOv11n as the baseline, three self-developed modules were progressively introduced: the lightweight detection head (LDHead), the WaveletPool module, and the ContextGuided semantic enhancement unit. Each module was designed to reduce computational cost and model parameters while maintaining or improving accuracy.

LDHead reduced parameters from 2.58 M to 2.46 M, slightly increasing mAP50-95 from 72.6% to 73.6%, demonstrating improved structural efficiency. WaveletPool further reduced GFLOPs from 6.3 to 5.4 and parameters to 2.16 M, with mAP50 rising to 95.8%, confirming its efficiency in extracting multi-scale spatial features. Although lightweight, the ContextGuided module consistently enhances semantic understanding and, when combined with other modules, provides more stable performance gains.

The final LCW-YOLO-n model, integrating all modules, achieves 96.4% mAP50 and 73.6% mAP50-95 with only 1.64 M parameters and 4.5 GFLOPs. These results demonstrate that carefully designed lightweight modules can reduce computational complexity without sacrificing performance, highlighting their strong potential for real-world deployment.

### 4.4. Comparative Analysis of C3K2 Modules Improved Based on CGBlock

To further assess the efficacy of the proposed ContextGuided (CG) module in enhancing the C3K2 structure, the module was integrated into the YOLOv11 framework and systematically compared with several mainstream lightweight feature enhancement strategies, including Faster, RVB, HDRAB, MambaOut, Star, and AP. All experiments were carried out on the private dataset’s test set to reflect complex and realistic detection scenarios, evaluating both detection performance and computational efficiency (Table 3).

The results indicate that the CG module significantly improves semantic representation and localization precision, achieving 95.2% mAP-50 and 71.2% mAP-50-95, outperforming most compared methods. Although AP attains a slightly higher mAP-50 (95.5%), it incurs a higher computational cost (2.4 M parameters, 6.3 GFLOPs, 4.9 MB model size). In contrast, the CG module reduces parameters to 2.2 M, GFLOPs to 5.4, and model size to 4.5 MB while sacrificing only 0.3% in mAP-50, demonstrating a favorable balance between model efficiency and accuracy.

In comparison with MambaOut, the CG module improves mAP-50-95 by 3.1 percentage points (71.2% vs. 68.1%) while simultaneously reducing computational complexity (GFLOPs were reduced by 1.5; parameters were reduced by 0.3 M), indicating clear advantages over alternative lightweight enhancement strategies.

The C3K2 module, characterized by moderate spatial resolution and rich channel depth, was chosen as the optimal insertion point for CGBlock. This placement enables the full exploitation of local channel interactions and integration of global contextual information, enhancing feature discriminability for small objects and improving robustness in complex backgrounds. CGBlock’s dynamic channel-guided mechanism adaptively allocates weights to different channels, effectively emphasizing informative channels while suppressing irrelevant or noisy signals. Consequently, the integration of CGBlock not only improves classification and localization performance but also maintains the lightweight nature of the network.

In summary, the ContextGuided module achieves a well-balanced integration of detection accuracy, computational efficiency, and model compactness. Its ability to enhance both local and global feature representation, coupled with low latency and memory footprint, confirms its aptitude for deployment in resource-limited environments, including embedded systems or UAV-based detection platforms.

### 4.5. Comparative Experiments for Sampling Layer Improvement

To rigorously evaluate the effectiveness of the proposed WaveletPool module in feature sampling, we integrated it into the YOLOv11 framework and compared it against both the baseline downsampling strategy and several representative alternatives, including ContextGuidedDown, SPDConv, PSConv, EUCB, and SRFD, under a unified experimental setup. The results are reported in Table 4.

WaveletPool achieves the highest Precision (97.3%), Recall (90.6%), and mAP50 (95.8%) while also maintaining a competitive mAP50-95 of 72.8%. In terms of efficiency, WaveletPool requires only 5.4 GFLOPs and 2.2 M parameters, resulting in a compact 4.5 MB model size. Compared with SPDConv, which provides strong accuracy at the cost of 11.3 GFLOPs, 4.6 M parameters, and 9.1 MB, WaveletPool reduces both computational and memory demands by more than half while preserving superior accuracy, thereby offering a favorable accuracy–efficiency trade-off. Although SRFD slightly surpasses WaveletPool in mAP50-95 (74.6%), its higher GFLOPs (7.6) and larger model size (5.2 MB) limit its suitability for resource-constrained environments.

These findings demonstrate that WaveletPool effectively replaces conventional strided convolution or pooling with a multi-scale, edge-preserving downsampling strategy that simultaneously enhances high-frequency sensitivity and reduces redundancy, making it particularly advantageous for lightweight deployment. In the following section, we extend the analysis to LDHead, focusing on its multi-task learning design and its contribution to improving localization accuracy.

Figure 8 illustrates the visualization of feature responses with and without Wavelet Pooling. Panel (a) shows the original images, where cigarette butts are embedded in complex backgrounds such as leaves, twigs, and stones. Panel (b) presents the heatmaps generated without Wavelet Pooling, where the model detects the targets but produces dispersed activations with blurred edges and noticeable background interference. Panel (c), after introducing Wavelet Pooling, exhibits more concentrated responses around the targets with sharper boundary contours while suppressing irrelevant background activations. These results confirm that Wavelet Pooling preserves high-frequency details during downsampling, thereby enhancing the edge-aware features of small objects and improving robustness and detection accuracy in cluttered natural environments.

### 4.6. Comparative Experiments on Detection Head Improvement

To verify the performance of the proposed LDHead lightweight detection head, it was compared with several mainstream lightweight detection head architectures, including SEAMHead, EfficientHead, and LSCD. Evaluation metrics included Precision, Recall, mAP values, number of parameters, GFLOPs, and model size. The results are presented in Table 5.

The results indicate that LDHead outperforms all compared methods, achieving the highest mAP-50 (95.0%), mAP-50-95 (73.6%), and Recall (91.2%) while maintaining a reasonable model size of 5.0 MB and a parameter count of 2.5 M, demonstrating superior target recognition capability. Although EfficientHead has slightly fewer parameters (2.3 M), LDHead achieves a 0.4% improvement in mAP-50-95 and a 4.7% increase in Recall, showing that its design better preserves detection performance without compromising lightweight requirements.

LDHead has 6.2 GFLOPs, slightly higher than that of other methods (5.1–5.8), yet remains within a practical range for deployment. This trade-off between lightweight design and enhanced robustness and detection accuracy highlights LDHead’s potential for real-world applications in embedded and edge computing environments. Importantly, LDHead’s multi-task fusion mechanism—jointly predicting classification, bounding box regression, segmentation, and pose—enhances the perception of small objects, edge details, and local structures at minimal computational cost, supporting fine-grained detection in complex scenarios.

### 4.7. Comparative Experiments with YOLO Series Models

To validate the overall performance of LCW-YOLO-n, a comparison was conducted against different versions of the YOLO series, including YOLOv8n and YOLOv11-n. The VisDrone2019 validation set was used as the experimental dataset to evaluate model performance under complex real-world conditions. Evaluation metrics included Precision, Recall, mAP50, mAP50-95, number of parameters, GFLOPs, and inference speed (ms). The results are summarized in Table 6.

LCW-YOLO-n significantly outperforms all compared models in terms of mAP50 (49.0%) and Precision (71.4%). Despite having a lightweight design with only 1.6 M parameters and 4.5 GFLOPs, it achieves higher detection accuracy than YOLOv8-n (43.1%) and YOLOv9-t (44.9%). Although its inference speed is 3.6 ms—slightly slower than YOLOv8-n (2.5 ms) and YOLOv11-n (2.4 ms)—LCW-YOLO-n strikes a superior balance between enhanced accuracy and computational efficiency.

While YOLOv9-t attains a marginally higher Recall (43.3%), its GFLOPs reach 10.7, and its Precision is inferior to that of LCW-YOLO-n. This demonstrates that LCW-YOLO-n offers better overall performance with a lightweight structure, rendering it especially well-suited for deployment on resource-limited edge devices and real-time detection applications.

### 4.8. Comparative Experiments with Classical Algorithms

To assess the overall performance of LCW-YOLO-n in lightweight target detection, we systematically compared it with several classical detection algorithms, including two-stage methods (Faster R-CNN, Cascade R-CNN), single-stage methods (RetinaNet, EfficientDet, ATSS), novel architectures (DTSSNet), and the improved YOLO variant SPDC-YOLO-n. The experimental results are summarized in Table 7.

Regarding detection accuracy, LCW-YOLO-n achieves an mAP50 of 49.0%, outperforming many classical methods such as Cascade R-CNN (39.3%), EfficientDet (38.5%), and DTSSNet (39.9%). Although its mAP50-95 of 26.6% is slightly lower than that of SPDC-YOLO-n (28.4%), LCW-YOLO-n maintains an extremely lightweight design with only 1.6 M parameters, substantially fewer than that of EfficientDet (34.7 M) and Cascade R-CNN (68.9 M). Furthermore, with 4.5 GFLOPs and an inference speed of 3.6 ms, LCW-YOLO-n significantly surpasses conventional detectors in computational efficiency.

In stark contrast, Faster R-CNN and RetinaNet incur heavy computational overheads of 208 and 210 GFLOPs, with inference speeds of 72.1 ms and 74.3 ms, respectively, making them unsuitable for real-time detection demands. Consequently, LCW-YOLO-n not only remains competitive in detection accuracy but also demonstrates clear advantages in lightweight deployment and enables real-time inference, which makes it well-suited for edge devices and environments with constrained computational resources.

### 4.9. Comparative Experiments with Different Datasets

To further assess the generality and robustness of the proposed LCW-YOLO-n model across various scenarios, experiments were carried out on three representative datasets: ForestButt and Cigarette Butts for complex detection tasks and VisDrone2019 for challenging small-target dense scenes. The experimental results are presented in Table 8.

On the ForestButt dataset, LCW-YOLO-n achieves an mAP50 of 96.4%, which is 2.1% higher than that of YOLOv11-n, with an improvement of 1.0% in mAP50-95. Additionally, both Precision and Recall are improved, while the number of parameters is reduced by 1.0 M and GFLOPs by 1.8. This demonstrates the model’s strong representational capability in forest foreign object detection tasks.

For the cigarette butt detection task, LCW-YOLO-n attains an mAP50 of 87.2%, surpassing YOLOv11-n by 2%, and maintains a comparable mAP50-95. The model additionally demonstrates substantial reductions in both parameter count and computational complexity, indicating an efficient structure with strong generalization ability.

In the VisDrone2019 small target detection task, LCW-YOLO-n achieves an mAP50 of 49.0%, marking a 6.5% improvement over YOLOv11-n. Meanwhile, computational complexity and model size are greatly reduced, underscoring the model’s adaptability to resource-constrained devices and real-time detection scenarios.

In summary, LCW-YOLO-n demonstrates outstanding detection performance and lightweight advantages across multiple datasets, highlighting its strong potential for practical deployment.

As shown in Figure 9, the image was captured on a forest floor covered by a complex background composed of dead grass, branches, and fallen leaves. In such a natural environment, cigarette butt targets tend to be small and closely match the background color, making them easily obscured or camouflaged. This challenge is further exacerbated when targets appear at multiple scales, significantly reducing detection performance. The cigarette butts detected by our proposed method are marked with blue bounding boxes in the detection results on the left, while the detection results of YOLOv11 are presented on the right with corresponding annotations. It is evident that YOLOv11 frequently misses cigarette butts that have fuzzy edges or colors similar to the background, whereas our method more accurately locates these targets by leveraging multi-layer feature fusion and contextual information compensation.

Figure 10 presents another natural ground scene, containing an even more cluttered background with additional distracting objects such as fallen leaves, branches, and rocks. This increased complexity further facilitates the camouflage of cigarette butts within the background. The detected cigarette butt locations from both models are highlighted with blue boxes. YOLOv11 demonstrates notable under-detection in this complex environment, failing to recognize several true cigarette butt targets. In contrast, our method significantly reduces missed detections, providing more comprehensive target localization and demonstrating the superior suppression of under-reporting. Moreover, across varying target positions and scene conditions, our model consistently captures the subtle edge information separating the cigarette butts from the background. Whether the target is located near the image boundary, amidst severe background clutter, or under uneven illumination and camouflage, our model responds effectively. This sensitivity to edge details arises from its robust multi-scale feature extraction and contextual semantic enhancement, which greatly improve its robustness and adaptability in challenging natural environments.

As shown in Figure 11, the heatmap visualization reveals that LCW-YOLO exhibits a more concentrated and pronounced response within the cigarette butt target regions, indicating that its attention mechanism effectively focuses on the critical areas of the target. In contrast, the heatmap of YOLOv11 shows a more diffuse and weaker response in key regions, which is more susceptible to interference from complex backgrounds, resulting in reduced detection accuracy. LCW-YOLO enhances the perception of high-frequency details through the integration of the Wavelet Pooling module and improves the depth and precision of feature representation with the ContextGuided (CGBlock) module, enabling the model to be more selective and discriminative in the spatial distribution of attention. Additionally, the LDHead detection head is structurally redesigned to guide the model towards a more stable and consistent focusing strategy in multi-task scenarios, further strengthening its responsiveness to true cigarette butt targets. Overall, these improvements significantly enhance the robustness of LCW-YOLO in handling complex natural backgrounds such as woodland, grass, and gravel, resulting in stronger anti-interference capability and improved generalization performance.

## 5. Discussion

In this study, we introduce a lightweight target detection framework built upon YOLOv11 that simultaneously balances detection accuracy and model efficiency. Our approach focuses on three key innovations: the introduction of the Wavelet Pooling module to enhance edge retention during downsampling; the integration of the Context-Guided Block (CGBlock) within the backbone to enhance contextual understanding; and the design of the LDHead detection head, which markedly reduces parameters and computational complexity while facilitating scale-specific feature modeling. The experimental results indicate that the proposed approach surpasses the original YOLOv11 in multi-class target detection, particularly excelling in small-object recognition and complex background scenarios.

It is important to emphasize that the lightweighting strategy employed in this work does not consist of merely a reduction in network size but rather an advancement achieved through synergistic structural reconfiguration and information enhancement. For instance, Wavelet Pooling compresses spatial dimensions while preserving high-frequency detail information, effectively compensating for the edge feature loss common to traditional pooling operations. The CGBlock module models both local and global contexts via a multipath fusion mechanism, substantially enhancing the semantic representation capability of features. Furthermore, the LDHead detection head combines a scale-guided strategy with depth-separable convolutions, dramatically reducing FLOPs and parameter redundancy without sacrificing multi-scale detection capability.

From a deployment perspective, the proposed model exhibits strong engineering adaptability. Its performance on embedded platforms and low-power devices validates its potential for real-world applications, especially in resource-constrained environments such as video surveillance in forested areas and industrial safety inspections.

Despite achieving a favorable trade-off between accuracy and efficiency, there is still potential for further optimization. Future work will explore advanced model compression techniques, including pruning strategies to eliminate non-essential parameter pathways and the incorporation of knowledge distillation to develop smaller student models while maintaining accuracy. Additionally, tailoring convolutional operators and scheduling mechanisms for specific embedded platforms will be pursued to further enhance deployment efficiency.

## 6. Conclusions

In the present study, a lightweight improvement framework based on YOLOv11 is proposed, and Wavelet Pooling, a CGBlock-enhanced C3K2 module, and a lightweight LDHead detection head are designed from three key modules: a feature downsampling strategy, context modeling mechanism, and detection head structure, respectively. Experiments show that the model significantly reduces computational complexity and the number of model parameters without substantially compromising detection accuracy and exhibits excellent deployment capability and real-time performance.

This study provides a solution that combines accuracy, efficiency, and flexibility for the target detection task in resource-constrained scenarios, which is particularly suitable for applications demanding high real-time performance, such as UAV inspection, mobile monitoring, and forest fire smoke detection. Future work will focus on further lightweighting techniques, including structural pruning, knowledge distillation, and neural architecture search, combined with platform-specific optimization strategies, to enhance the model’s scalability and deployment versatility, thereby providing technical support for the development of edge intelligent sensing systems.

## Figures and Tables

**Figure 1 sensors-25-06209-f001:**
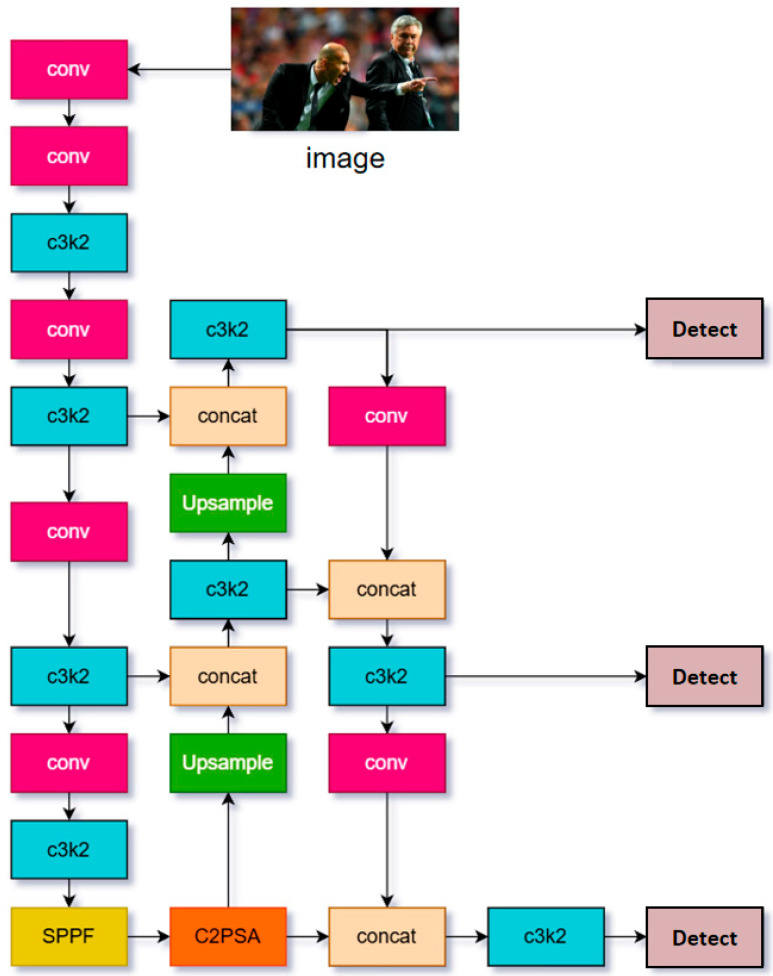
YOLOv11 structure diagram. Different colors indicate different network structures, arrows represent the transmission and fusion of features between layers, and the Detect modules denote the outputs of multi-scale detection results. The same color and symbol definitions apply to subsequent figures and are not repeated.

**Figure 2 sensors-25-06209-f002:**
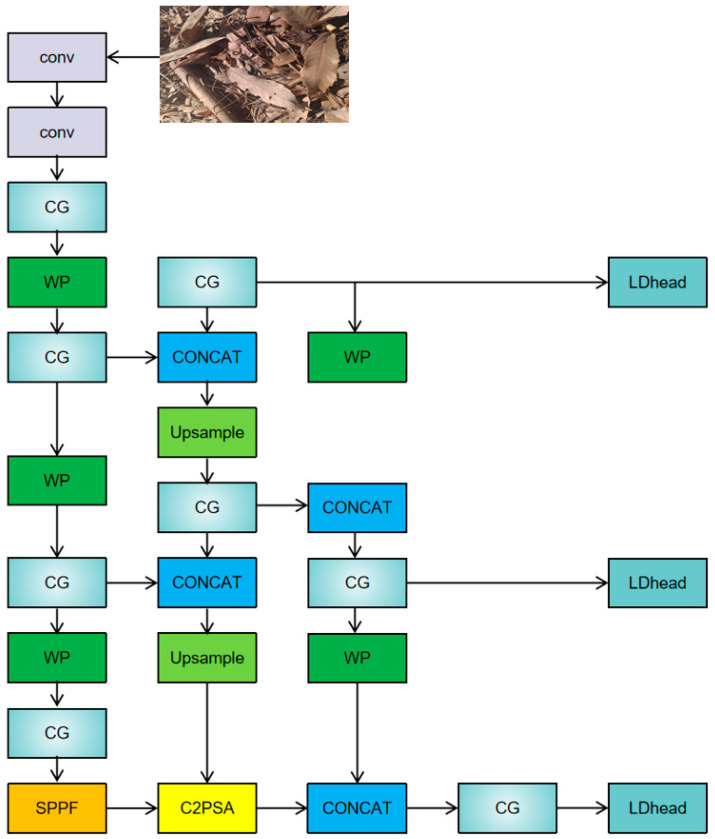
Structure diagram of LCW-YOLO and its differences from YOLOv11.

**Figure 3 sensors-25-06209-f003:**
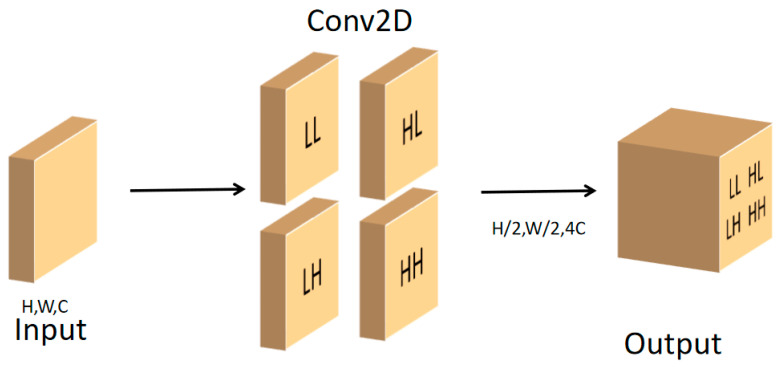
Structure diagram of Wavelet Pooling module.

**Figure 4 sensors-25-06209-f004:**
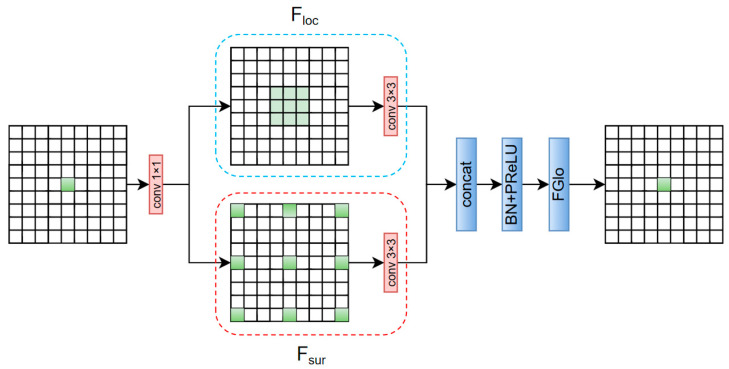
Structure diagram of CGBlock module.

**Figure 5 sensors-25-06209-f005:**
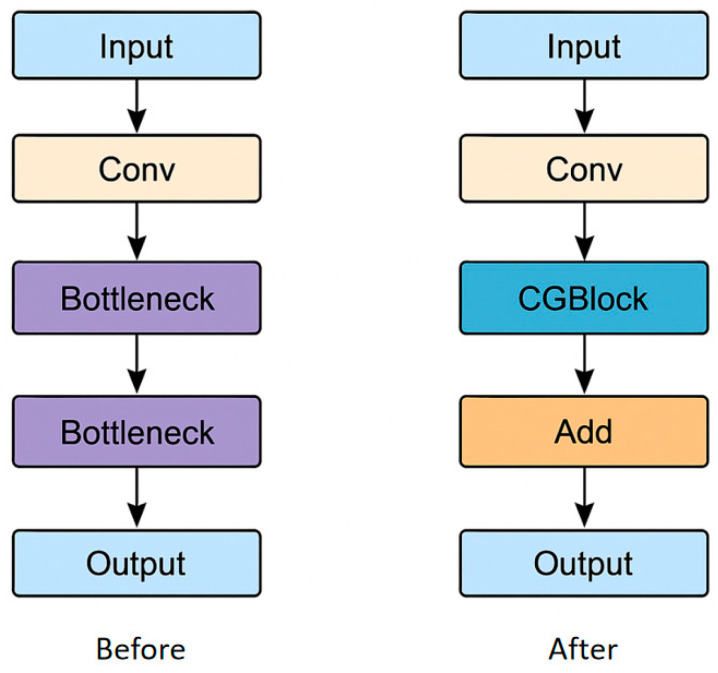
Comparison of network structures before and after adding CGBlock module.

**Figure 6 sensors-25-06209-f006:**
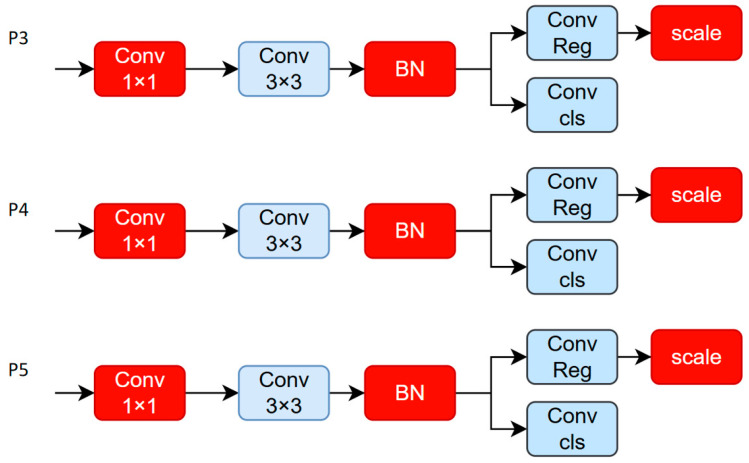
Structure diagram of LDHead.

**Figure 7 sensors-25-06209-f007:**
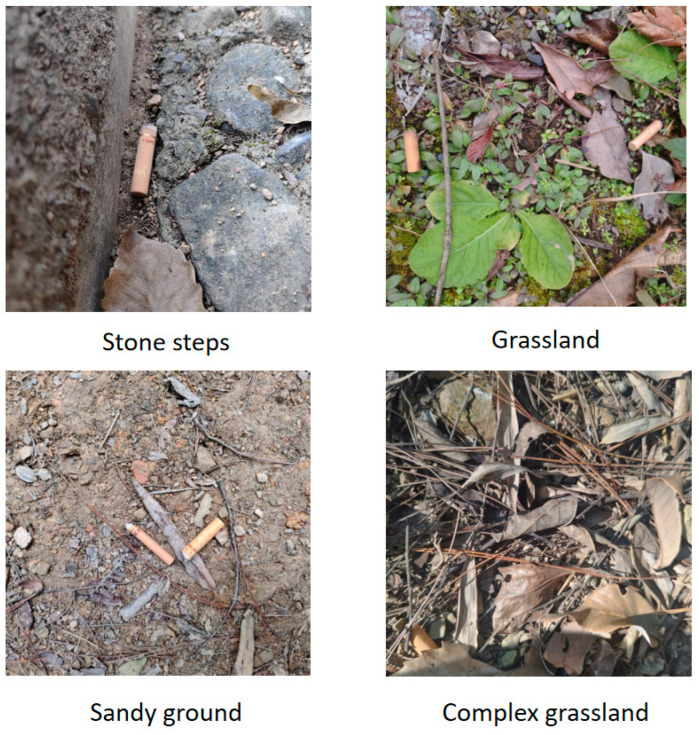
Detection targets across diverse environmental scenarios.

**Figure 8 sensors-25-06209-f008:**
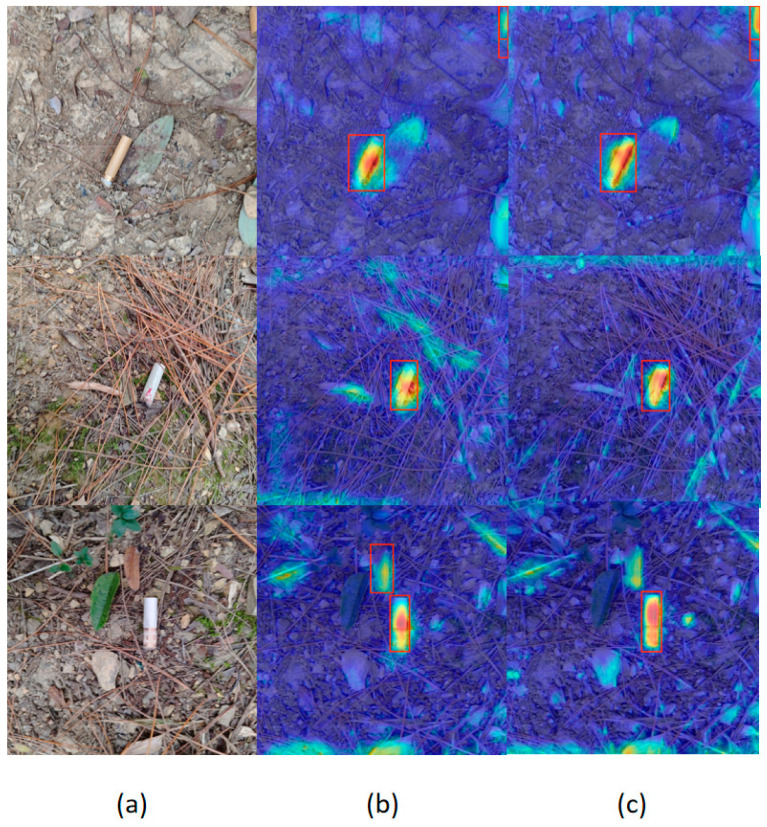
Visualization of feature response with and without Wavelet Pooling: (**a**) original image; (**b**) heatmap without Wavelet Pooling; (**c**) heatmap with Wavelet Pooling. In the heatmap, different color regions indicate the model’s attention level on the image. Red and yellow areas correspond to high attention (high probability of object presence), while blue areas indicate low attention. The red bounding boxes mark the locations of detected objects by the model. The subsequent heat maps are consistent.

**Figure 9 sensors-25-06209-f009:**
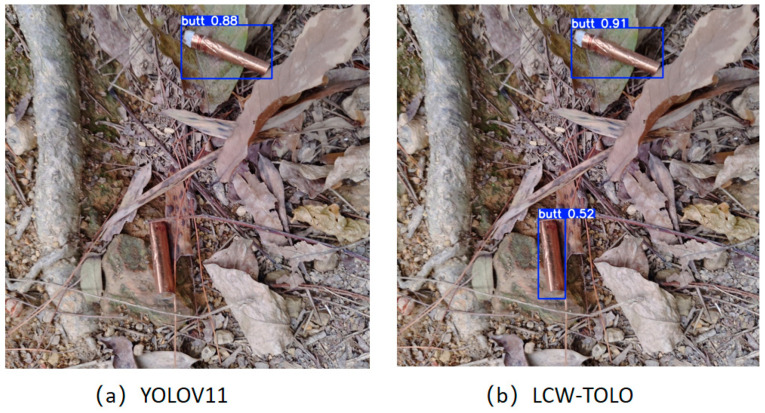
Images of mountain forest and grassland areas.

**Figure 10 sensors-25-06209-f010:**
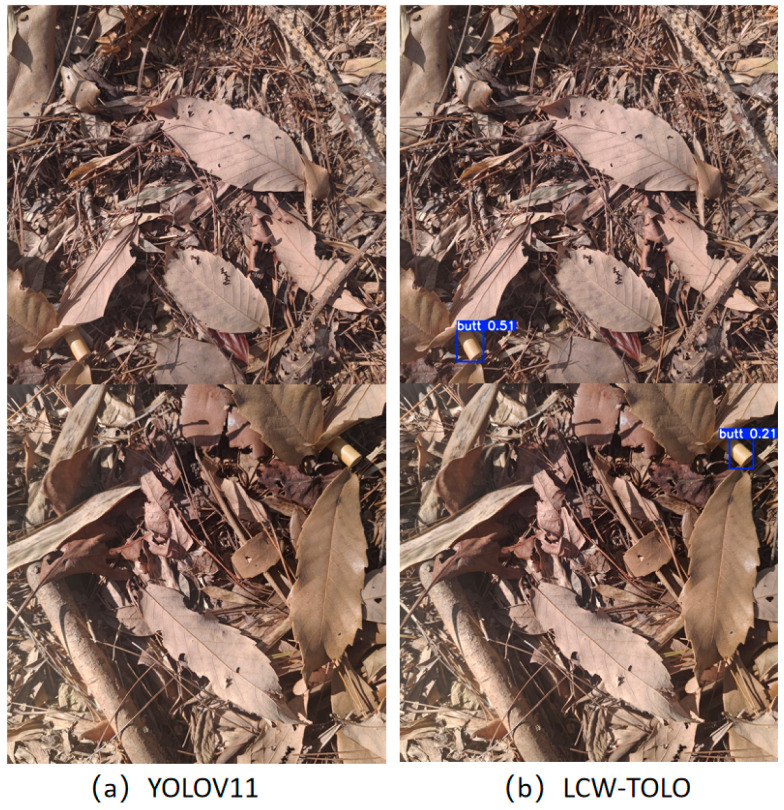
Images of more complex mountain forest and grassland areas.

**Figure 11 sensors-25-06209-f011:**
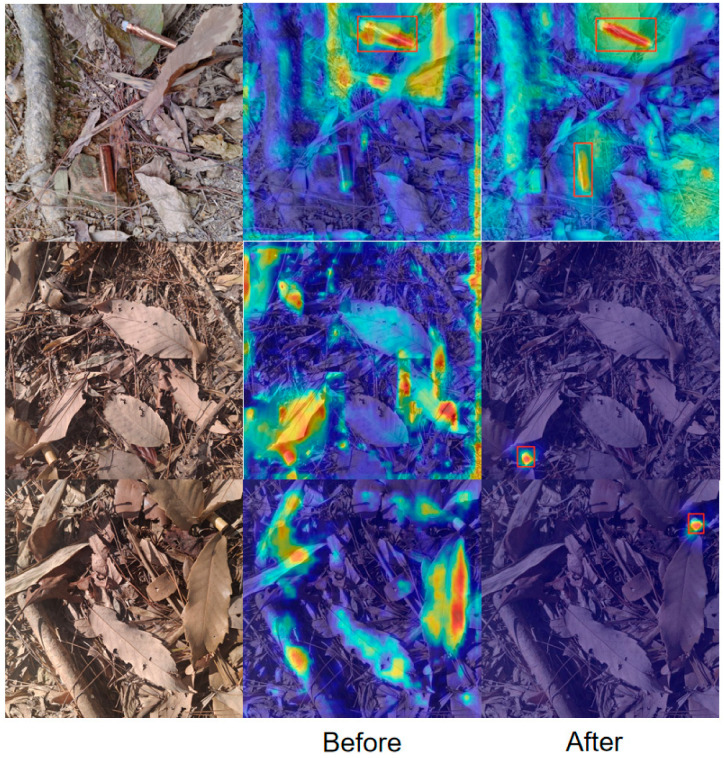
Comparative analysis of heatmaps.

**Table 1 sensors-25-06209-t001:** Configuration of experimental parameters.

**Training Parameters**	Batch size	32
	Epochs	300
	Optimizer	SGD
	Image size	640 × 640
	Momentum	0.937
	Initial learning rate	0.01
**Computer Configurations**	Operating system	Windows 11
	Graphics memory	8 GB

**Table 2 sensors-25-06209-t002:** Ablation experiment results of ForestButt. Bold values in the table indicate the best performance in each row/column. The same notation is applied to all subsequent tables.

	LDHead	WaveletPool	ContextGuided	P	R	mAP50	mAP50-95	Parameters (M)	GFLOPs
**YOLO11n**				97.0	88.8	94.3	72.6	2.6	6.3
	√			96.5	91.2	95	73.6	2.5	6.2
		√		97.3	90.6	95.8	72.8	2.2	5.4
			√	93.9	87.6	95.2	71.2	2.2	5.4
	√	√		95.4	90	94.1	**74.4**	2.0	5.3
	√		√	97.4	**91.8**	**97.4**	73.7	2.1	5.3
		√	√	95.1	88.8	92.7	71.4	1.8	4.6
**LCW-YOLO-n**	√	√	√	**99.4**	90.5	96.4	73.6	**1.6**	**4.5**

**Table 3 sensors-25-06209-t003:** Comparison of different C3K2 improvement methods.

	F1	P (%)	R (%)	mAP50 (%)	mAP50-95 (%)	Parameters (M)	GFLOPs	Model Size
**Faster**	91.1	**95.6**	87.1	92.9	69.6	2.3	5.8	4.7 MB
**RVB** [30]	91.2	95.1	87.7	94.5	70.6	2.3	5.8	4.7 MB
**HDRAB** [31]	89.2	91.4	87.1	93.4	71.7	2.6	6.6	5.3 MB
**MambaOut**	89.9	92.0	**87.9**	93.3	68.1	2.5	6.9	5.1 MB
**Star**	88.8	91.9	85.9	94.0	72.0	2.5	6.4	5.1 MB
**AP**	89.6	93.6	86.0	**95.5**	**73.4**	2.4	6.3	4.9 MB
**ContextGuided(ours)**	**92.9**	93.9	87.6	95.2	71.2	**2.2**	**5.4**	**4.5 MB**

**Table 4 sensors-25-06209-t004:** Comparison of different sampling layer methods.

	F1	P (%)	R (%)	mAP50 (%)	mAP50-95 (%)	Parameters	GFLOPs	Model Size
**ContextGuidedDown**	89.6	94.3	85.3	92.7	70.8	3.5	9.0	7.1 MB
**SPDConv** [32]	91.4	95.5	87.7	93.4	71.8	4.6	11.3	9.1 MB
**PSConv** [33]	89.4	94.0	85.3	92.5	70.3	2.5	6.3	5.0 MB
**EUCB** [34]	91.2	93.1	89.4	94.1	72.3	2.7	6.8	5.4 MB
**SRFD** [35]	90.0	91.9	88.2	94.1	**74.6**	2.6	7.6	5.2 MB
**WaveletPool(ours)**	**92.5**	**97.3**	**90.6**	**95.8**	72.8	**2.2**	**5.4**	**4.5 MB**

**Table 5 sensors-25-06209-t005:** Comparison of LDHead with other detection heads.

	F1	P (%)	R (%)	mAP50 (%)	mAP50-95 (%)	Parameters	GFLOPs	Model Size
**SEAMHead** [36]	90.2	**97.3**	84.1	94.6	73.4	2.5	5.8	5.1 MB
**EfficientHead**	90.5	94.8	86.5	94.7	73.2	**2.3**	**5.1**	**4.7 MB**
**LSCD**	90.2	91.6	88.8	94.7	72.3	2.4	5.6	4.9 MB
**LDHead(ours)**	**93.8**	96.5	**91.2**	**95.0**	**73.6**	2.5	6.2	5.0 MB

**Table 6 sensors-25-06209-t006:** Comparison of YOLO (You Only Look Once) versions.

	P (%)	R (%)	mAP50 (%)	mAP50-95 (%)	Parameters	GFLOPs	Test Speed (ms)
**Yolov8-n**	51.5	42.3	43.1	26.0	3.0	8.1	2.5
**Yolov9-t**	54.3	**43.3**	44.9	22.7	2.6	10.7	3.8
**Yolov10-n** [34]	51.4	40.5	42.2	25.6	2.7	8.2	3.0
**Yolov11-n**	50.5	42.2	42.5	25.8	2.6	6.3	**2.4**
**LCW-YOLOn(ours)**	**71.4**	42.4	**49.0**	**26.6**	**1.6**	**4.5**	3.6

**Table 7 sensors-25-06209-t007:** Comparison of LCW-YOLO with classic models.

	mAP50 (%)	mAP50-95 (%)	Parameters	GFLOPs	Test Speed (ms)
**Faster R-CNN**	33.2	19.9	-	208.0	72.1
**RetinaNet** [37]	29.0	17.2	-	210.0	74.3
**Cascade R-CNN**	39.3	25.6	68.9	-	-
**EfficientDet** [38]	38.5	24.6	34.7	-	-
**DTSSNet** [39]	39.9	24.2	10.1	50.4	12.8
**ATSS** [40]	31.7	18.6	10.3	57.0	13.2
**SPDC-YOLO-n** [41]	46.5	**28.4**	2.0	10.0	**3.5**
**LCW-YOLOn(ours)**	**49.0**	26.6	**1.6**	**4.5**	3.6

**Table 8 sensors-25-06209-t008:** Comparison of LCW-YOLO and YOLOv11 on different datasets.

Dataset	Model	P (%)	R (%)	mAP50 (%)	mAP50–95 (%)	Params	GFLOPs	Model Size
**ForestButt**	**Yolov11-n**	97	88.8	94.3	72.6	2.6	6.3	5.3 MB
	**LCW-YOLOn (ours)**	**99.4**	**90.5**	**96.4**	**73.6**	**1.6**	**4.5**	**3.5 MB**
**Cigarette Butts**	**Yolov11-n**	**88.7**	78.8	85.2	**58.6**	2.6	6.3	5.3 MB
	**LCW-YOLOn (ours)**	85.5	**79.1**	**87.2**	58.5	**1.6**	**4.5**	**3.5 MB**
**VisDrone2019**	**Yolov11-n**	50.5	42.2	42.5	25.8	2.6	6.3	5.2 MB
	**LCW-YOLOn (ours)**	**71.4**	**42.4**	**49**	**26.6**	**1.6**	**4.5**	**3.6 MB**

## Data Availability

Data is contained within the article.

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
