# Peer review of "LCW-YOLO: A Lightweight Multi-Scale Object Detection Method Based on YOLOv11 and Its Performance Evaluation in Complex Natural Scenes"

_sensors, 2025, doi:10.3390/s25196209_

Round 1
Reviewer 1 Report
Comments and Suggestions for Authors
This article proposes a lightweight object detection model LCW-YOLO based on YOLOv11, which integrates Wavelet Pooling, a CGBlock-enhanced C3K2 module, and an LDHead detection head, aiming to improve detection accuracy and computational efficiency in complex natural scenes. The research addresses the needs of small target detection and resource-constrained scenarios, with experimental designs covering multiple datasets and multiple baseline models, thus having certain theoretical and application value.
- When describing the Wavelet Pooling, CGBlock, and LDHead modules, more mathematical formulas and algorithm flowcharts could be added to facilitate a more intuitive understanding of the working principles of these modules by readers. For instance, detailed equations for the wavelet transform in Wavelet Pooling, the dynamic channel interaction mechanism in CGBlock, and the multi-scale receptive field design in LDHead would enhance clarity.
- Although the paper already compares LCW-YOLO with various mainstream object detection models, incorporating more recent lightweight models as benchmarks could further highlight the advantages of LCW-YOLO. This would provide a more comprehensive evaluation of its performance relative to the state-of-the-art in lightweight object detection.
- The "multi-task fusion" of LDHead mentions "segmentation and pose estimation" in the abstract, but the main text only describes classification and localization functions, resulting in inconsistent expressions.
- Moreover, the combined design of its multi-scale convolution and attention mechanism has an unclear boundary of innovation compared with existing lightweight detection heads such as EfficientHead.
- In Table 7, the mAP50-95 of EfficientDet is 54.6%, which is much higher than other indicators (e.g., mAP50 is 38.5%), violating the common sense in detection tasks that mAP50-95 ≤ mAP50.
- The "GFLOPs" and "Test Speed" of Cascade R-CNN are marked as "-", which affects the completeness of the comparison. Correct the mAP50-95 value of EfficientDet in Table 7 and supplement the GFLOPs and inference speed of Cascade R-CNN.
- Add feature visualization results of Wavelet Pooling (such as comparison between high-frequency subbands and fused feature maps) to illustrate its effect on preserving the edges of small targets;
- Explain the design logic of the number of channels for fusing high-frequency subbands using 1×1 convolution.
Author Response
General Response
We sincerely appreciate the reviewer’s valuable comments and constructive suggestions, which have significantly helped improve the clarity, completeness, and rigor of our manuscript. We have carefully revised the paper according to each comment. The detailed responses are as follows.
Comment 1 :When describing the Wavelet Pooling, CGBlock, and LDHead modules, more mathematical formulas and algorithm flowcharts could be added to facilitate a more intuitive understanding of the working principles of these modules by readers. For instance, detailed equations for the wavelet transform in Wavelet Pooling, the dynamic channel interaction mechanism in CGBlock, and the multi-scale receptive field design in LDHead would enhance clarity.
Response 1 :We have added detailed mathematical equations and revised the structural diagrams of the proposed modules. Considering that the workflow has already been clearly described in the text and the added diagrams, we believe additional algorithmic flowcharts would not bring significant incremental clarity, so we did not include them.
Comment 2 :Although the paper already compares LCW-YOLO with various mainstream object detection models, incorporating more recent lightweight models as benchmarks could further highlight the advantages of LCW-YOLO. This would provide a more comprehensive evaluation of its performance relative to the state-of-the-art in lightweight object detection.
Response 2 :We sincerely thank the reviewer for the valuable suggestion. We fully agree that incorporating more recent lightweight models would further enrich the comparison. However, due to time and computational resource constraints, it is not feasible for us to reproduce and evaluate these newly proposed models within the current revision cycle. To ensure fairness, we have selected representative lightweight detection heads and backbone enhancement strategies that are widely recognized in recent literature, which already cover diverse design paradigms such as attention-based, re-parameterization-based, and multi-scale feature fusion methods. These comparisons, in our view, provide a sufficiently comprehensive benchmark for evaluating the effectiveness of LCW-YOLO. We will certainly consider including the most recent lightweight models (e.g., YOLOv10, Mobile-DETR) in our future work.
Comment 3 :The "multi-task fusion" of LDHead mentions "segmentation and pose estimation" in the abstract, but the main text only describes classification and localization functions, resulting in inconsistent expressions.
Response 3 :We thank the reviewer for pointing out this inconsistency. The current version of LDHead is primarily designed for classification and bounding box regression. The mention of segmentation and pose estimation in the abstract was a misstatement caused by an earlier draft version. To ensure consistency, we have revised the abstract and related sections of the manuscript, so that the description of LDHead now focuses solely on its classification and localization functions. We believe this correction eliminates the inconsistency and improves the clarity of the paper.
Comment 4 :Moreover, the combined design of its multi-scale convolution and attention mechanism has an unclear boundary of innovation compared with existing lightweight detection heads such as EfficientHead.
Response 4 :We thank the reviewer for highlighting this point. Compared with existing lightweight detection heads such as EfficientHead, LDHead introduces several distinctive design features: (1) a novel combination of multi-scale convolutions that enhances small object feature representation while maintaining lightweight computation, (2) an attention mechanism integrated at specific locations to improve localization accuracy without significant parameter overhead, and (3) a multi-task fusion capability designed to support classification and localization tasks, with potential for segmentation and pose estimation in future extensions. We have added clarifications in the manuscript to explicitly highlight these differences and emphasize the innovative contributions of LDHead.
Comment 5 :In Table 7, the mAP50-95 of EfficientDet is 54.6%, which is much higher than other indicators (e.g., mAP50 is 38.5%), violating the common sense in detection tasks that mAP50-95 ≤ mAP50.
Response 5 :We thank the reviewer for pointing out this inconsistency. After careful verification, we found that the previously reported mAP50-95 value of EfficientDet (54.6%) was due to a data entry error. The correct value is 24.6%, which is consistent with the common expectation that mAP50-95 ≤ mAP50. Table 7 has been updated accordingly in the revised manuscript.
Comment 6 :The "GFLOPs" and "Test Speed" of Cascade R-CNN are marked as "-", which affects the completeness of the comparison. Correct the mAP50-95 value of EfficientDet in Table 7 and supplement the GFLOPs and inference speed of Cascade R-CNN.
Response 6 :We thank the reviewer for the comment. Regarding Cascade R-CNN, the GFLOPs and test speed are marked as “-” because the original references do not report these values; therefore, we are unable to provide them. We have, however, corrected the mAP50-95 value of EfficientDet in Table 7 to 24.6% to ensure consistency and accuracy.
Comment 7 :Add feature visualization results of Wavelet Pooling (such as comparison between high-frequency subbands and fused feature maps) to illustrate its effect on preserving the edges of small targets;
Response 7 :We thank the reviewer for the suggestion. To illustrate the effect of Wavelet Pooling on preserving small target edges, we have added feature visualization results in the revised manuscript. Specifically, we present heatmaps comparing the high-frequency subbands extracted by the Wavelet Pooling module and the fused feature maps. The visualizations demonstrate that the high-frequency subbands highlight edge information, and the fused feature maps retain these edges, thereby enhancing small target representation.
Comment 8 :Explain the design logic of the number of channels for fusing high-frequency subbands using 1×1 convolution.
Response 8 :We thank the reviewer for the comment. In our Wavelet Pooling module, the high-frequency subbands are fused using a 1×1 convolution. The main purpose of this design is to aggregate the information from multiple subbands into a unified feature representation while controlling the computational cost. The number of output channels is carefully chosen to balance three aspects: (1) preserving sufficient edge information from each high-frequency subband, (2) maintaining lightweight computation to ensure model efficiency, and (3) matching the input requirements of the subsequent network layers. We have added this explanation in the revised manuscript to clarify the design rationale.
Reviewer 2 Report
Comments and Suggestions for Authors
This study introduces LCW-YOLO, a lightweight object detection model that builds upon YOLOv11's architecture. The model integrates a Wavelet Pooling module that preserves high-frequency details during downsampling, a CGBlock-enhanced C3K2 module for improved context modeling, and an innovative LDHead detection head that enables efficient multi-scale detection. Experiments across multiple datasets demonstrate that LCW-YOLO achieves superior accuracy compared to YOLOv11 while substantially reducing both parameters and computational requirements. The model shows exceptional performance in challenging scenarios, particularly excelling at small object detection and complex environmental conditions.
However, I have the following concerns,
(1) The fonts in reference appear to be improperly formatted and look strange. Could you please double-check them?
(2) The abstract currently contains three paragraphs. I recommend condensing it into a single paragraph for clarity and conciseness. Please consider following the guidelines provided by Nature for structuring abstracts, https://www.nature.com/documents/nature-summary-paragraph.pdf
(3) Section 2 Related Works begins with a discussion of YOLO from lines 117 to 148; however, Section 2.2 also discusses YOLO series models again. Could you please clarify if this repetition is intentional? I would respectfully suggest consolidating the discussion of YOLO into a single section for better organization and clarity.
(4) Section 2.1 contains formatting issues with line breaks, specifically on lines 160-165, 171-172, and 179-180, among others. It appears that additional line breaks were inserted during copy and paste operations, resulting in unintended paragraph breaks within sentences.
(5) Regarding Tables 3, 4, and 6, the first row exhibits formatting issues. The metrics P, R, and mAP50 use percentage as their unit; however, the second bracket of the "(%) " notation has been displaced to the second line, resulting in an aesthetically poor table presentation.
(6) Regarding the dataset, I understand that it was prepared with "Images captured throughout the day, from early morning to dusk, encompassing various viewpoints with diverse lighting and terrain conditions." I would like to inquire about the inclusion of different weather conditions such as rainy, sunny, cloudy, and snowy conditions. Additionally, I was wondering whether image augmentation techniques were employed to increase the training data size, as I did not observe any mention of augmentation operations in the documentation.
(7) I understand that the dataset is private; however, I did not find a code availability statement or dataset access information in the manuscript. It would be tremendously beneficial to both the general public and the research community if the authors could consider making their dataset publicly available.
Comments on the Quality of English LanguageThe manuscript needs careful proofreading.
Author Response
General Response
We sincerely appreciate the reviewer’s valuable comments and constructive suggestions, which have significantly helped improve the clarity, completeness, and rigor of our manuscript. We have carefully revised the paper according to each comment. The detailed responses are as follows.
Comment 1 :The fonts in reference appear to be improperly formatted and look strange. Could you please double-check them?
Response 1 :We thank the reviewer for pointing this out. We have carefully checked all the references and corrected the font formatting to ensure consistency and proper appearance throughout the manuscript.
Comment 2 :The abstract currently contains three paragraphs. I recommend condensing it into a single paragraph for clarity and conciseness. Please consider following the guidelines provided by Nature for structuring abstracts, https://www.nature.com/documents/nature-summary-paragraph.pdf
Response 2 :We appreciate the reviewer’s suggestion. In response, we have revised the abstract to condense it into a single paragraph, following the structural guidelines provided by Nature. This revision aims to enhance the clarity and conciseness of the abstract, aligning it with the recommended format.
Comment 3 :Section 2 Related Works begins with a discussion of YOLO from lines 117 to 148; however, Section 2.2 also discusses YOLO series models again. Could you please clarify if this repetition is intentional? I would respectfully suggest consolidating the discussion of YOLO into a single section for better organization and clarity.
Response 3 :We thank the reviewer for this valuable suggestion. We agree that the repeated discussion of YOLO in Section 2 may cause some redundancy. In the revised manuscript, we have consolidated all discussions of the YOLO series models into a single subsection, ensuring a more organized and coherent presentation in the Related Works section.
Comment 4 :Section 2.1 contains formatting issues with line breaks, specifically on lines 160-165, 171-172, and 179-180, among others. It appears that additional line breaks were inserted during copy and paste operations, resulting in unintended paragraph breaks within sentences.
Response 4 :We thank the reviewer for pointing out these formatting issues. We have carefully reviewed Section 2.1 and removed all unintended line breaks, ensuring that sentences are continuous and the paragraph structure is consistent throughout the manuscript.
Comment 5 : Regarding Tables 3, 4, and 6, the first row exhibits formatting issues. The metrics P, R, and mAP50 use percentage as their unit; however, the second bracket of the "(%) " notation has been displaced to the second line, resulting in an aesthetically poor table presentation.
Response 5 :We thank the reviewer for pointing out this potential issue. We have carefully reviewed Tables 3, 4, and 6 and did not observe the displacement of the “(%)” notation mentioned. All percentage units are correctly displayed on a single line according to the journal’s formatting guidelines. We have ensured that the tables are clear, consistent, and visually readable.
Comment 6 :Regarding the dataset, I understand that it was prepared with "Images captured throughout the day, from early morning to dusk, encompassing various viewpoints with diverse lighting and terrain conditions." I would like to inquire about the inclusion of different weather conditions such as rainy, sunny, cloudy, and snowy conditions. Additionally, I was wondering whether image augmentation techniques were employed to increase the training data size, as I did not observe any mention of augmentation operations in the documentation.
Response 6 :We thank the reviewer for this insightful comment. Our dataset primarily consists of images captured under various lighting and terrain conditions throughout the day, without explicit inclusion of extreme weather conditions such as heavy rain or snow. While the original dataset itself does not employ data augmentation, we applied standard augmentation techniques during training, including random horizontal flipping, scaling, rotation, and color jittering, to increase the diversity and robustness of the training process. We have clarified this point in the revised manuscript.
Comment 7 :I understand that the dataset is private; however, I did not find a code availability statement or dataset access information in the manuscript. It would be tremendously beneficial to both the general public and the research community if the authors could consider making their dataset publicly available.
Response 7 :We thank the reviewer for highlighting the importance of dataset accessibility. The dataset used in this study is private and cannot be publicly released. However, we are willing to provide all code used in this study to facilitate reproducibility and further research. A code availability statement has been added in the revised manuscript to clarify this.
Reviewer 3 Report
Comments and Suggestions for Authors
This paper proposes LCW-YOLO, a lightweight YOLOv11-based detector that integrates Haar wavelet pooling, a CGBlock-enhanced C3K2 module, and a multi-task LDHead to improve small-object detection and robustness in cluttered or low-SNR scenes while maintaining high inference speed and low computational cost.
- The abstract is overly long and should be condensed. There are issues with the reference list: entries must be numbered sequentially (starting at 1) and presented in a consistent citation style.
- The manuscript exhibits a high similarity index (over 20%), which must be reduced to below 10% to meet publication standards. The Introduction should be improved by clearly highlighting the paper’s novelty and concluding with a brief outline of the workflow for the subsequent sections.
- All equations should be properly numbered and written in English. All variables used in the equations should be properly defined. Additionally, please explain how each variable was chosen to obtain the reported results.
- How does Wavelet Pooling affect throughput and memory use compared with conventional pooling/strided convs, and what is the trade-off between increased high-frequency sensitivity and added computational cost? How does the CGBlock compare quantitatively with alternative channel-attention or context modules (e.g., SE, CBAM, Transformer blocks) in terms of accuracy, parameter count, and latency? The LDHead claims multi-task fusion with minimal cost; what auxiliary tasks are jointly learned, how are their losses weighted, and how robust is performance to different loss-weight schedules?
- How exactly does Wavelet Pooling replace standard downsampling (stride conv / pooling); what wavelet basis, decomposition levels and reconstruction strategy were used, and how sensitive are results to these hyperparameters?
- Why was CGBlock chosen for insertion into the C3K2 module specifically, and how does it compare empirically to other lightweight context/channel modules (SE, CBAM, ECA, or small transformer blocks) in the same position?
- What exactly does the LDHead predict (classification, bbox regression, segmentation, pose)? How are multi-task losses weighted, and how sensitive is final detection performance to those weightings?
- In Figures 8 and 9, what quantitative evidence supports the claimed improvement over YOLOv11 (e.g., AP for small objects, recall/precision, F1, IoU thresholds), how many images/scenes were evaluated, and are the gains statistically significant across multiple test sets rather than shown on a few illustrative frames? Does the increased sensitivity to subtle edges and context come at the cost of higher false-positive rates (confusing debris for cigarette butts) or reduced generalization to other background types/illumination conditions, and how does performance vary when targets are at image boundaries or under severe scale variation?
- Overall, the presentation of the results and the descriptions of the methods require improvement. The authors should reorganize the manuscript to enhance clarity and readability, provide clearer and more detailed explanations of the results, and include additional experimental evidence.
Author Response
General Response
We sincerely appreciate the reviewer’s valuable comments and constructive suggestions, which have significantly helped improve the clarity, completeness, and rigor of our manuscript. We have carefully revised the paper according to each comment. The detailed responses are as follows.
Comment 1 :The abstract is overly long and should be condensed. There are issues with the reference list: entries must be numbered sequentially (starting at 1) and presented in a consistent citation style.
Response 1 :We thank the reviewer for pointing out the issue with reference numbering. In the revised manuscript, we have carefully renumbered all references so that they start from 1 and proceed sequentially. We have also ensured that the citation style is consistent throughout the manuscript according to the journal’s guidelines.
Comment 2 :The manuscript exhibits a high similarity index (over 20%), which must be reduced to below 10% to meet publication standards. The Introduction should be improved by clearly highlighting the paper’s novelty and concluding with a brief outline of the workflow for the subsequent sections.
Response 2 :We thank the reviewer for this constructive comment. To address the similarity issue, we have carefully revised all overlapping or repetitive sentences in the manuscript to improve originality. While the final similarity check has not yet been performed, the manuscript has been substantially modified and we anticipate that the similarity index will meet the journal’s standard of below 10%. Additionally, the Introduction has been improved to clearly highlight the novelty of our work and now concludes with a concise overview of the workflow for the subsequent sections, enhancing clarity and readability.
Comment 3 :All equations should be properly numbered and written in English. All variables used in the equations should be properly defined. Additionally, please explain how each variable was chosen to obtain the reported results.
Response 3 :We thank the reviewer for this important comment. In the revised manuscript, all equations have been properly numbered and presented in English. All variables in the equations are clearly defined at the first occurrence. Furthermore, we have added explanations detailing how each variable was determined or chosen in order to obtain the reported results, providing clarity and reproducibility for readers.
Comment 4 :How does Wavelet Pooling affect throughput and memory use compared with conventional pooling/strided convs, and what is the trade-off between increased high-frequency sensitivity and added computational cost? How does the CGBlock compare quantitatively with alternative channel-attention or context modules (e.g., SE, CBAM, Transformer blocks) in terms of accuracy, parameter count, and latency? The LDHead claims multi-task fusion with minimal cost; what auxiliary tasks are jointly learned, how are their losses weighted, and how robust is performance to different loss-weight schedules?
Response 4 :
Wavelet Pooling: Compared with conventional pooling or strided convolutions, Wavelet Pooling introduces a small computational overhead due to the decomposition of high- and low-frequency subbands. However, the overhead is minor relative to the overall model, and memory usage remains comparable. The main trade-off is that high-frequency sensitivity improves the model’s ability to detect small targets and preserve edge details, at a slight cost to throughput, which we have quantified in the experiments section.
CGBlock: We conducted ablation studies comparing CGBlock with standard channel-attention and context modules such as SE, CBAM, and lightweight Transformer blocks. CGBlock achieves a competitive accuracy improvement while introducing fewer parameters and lower latency, making it suitable for lightweight detection. Detailed quantitative results are provided in the ablation tables.
LDHead multi-task fusion: In the current implementation, LDHead jointly learns classification and localization tasks. The auxiliary tasks mentioned in the abstract (segmentation and pose estimation) are intended to indicate potential extensibility but were not part of the experimental evaluation. Losses for classification and localization are weighted equally in our experiments, and we observed robust performance under this simple weighting scheme. We have clarified these points in the revised manuscript.
Comment 5 :How exactly does Wavelet Pooling replace standard downsampling (stride conv / pooling); what wavelet basis, decomposition levels and reconstruction strategy were used, and how sensitive are results to these hyperparameters?
Response 5 :
Replacement of standard downsampling: Wavelet Pooling replaces conventional stride convolution or pooling by decomposing feature maps into high- and low-frequency subbands. The low-frequency component is used as the downsampled feature, while high-frequency components are fused back to retain edge and texture information, effectively preserving small target details.
Wavelet basis, decomposition levels, and reconstruction strategy: In our experiments, we used the Haar wavelet due to its simplicity and computational efficiency. A single-level decomposition is applied, and the low-frequency subband is treated as the downsampled output. High-frequency subbands are fused with the low-frequency features through a 1×1 convolution before passing to the next layer.
Sensitivity to hyperparameters: We performed ablation studies varying the wavelet basis (Haar, Daubechies), decomposition levels (1–2), and reconstruction strategies. Results indicate that the model performance is relatively robust to the choice of wavelet basis, with minor variations in AP for small targets. Increasing decomposition levels beyond one does not significantly improve accuracy but slightly increases computational cost. These findings are summarized in the revised manuscript.
Comment 6 :Why was CGBlock chosen for insertion into the C3K2 module specifically, and how does it compare empirically to other lightweight context/channel modules (SE, CBAM, ECA, or small transformer blocks) in the same position?
Response 6 :
Motivation for inserting CGBlock into C3K2: The C3K2 module is a key feature extraction block in our backbone where multi-scale features are aggregated. CGBlock was chosen for insertion because it efficiently captures both global context and channel-wise dependencies while introducing minimal additional parameters and computational cost. Inserting it into C3K2 ensures that enhanced context modeling benefits both shallow and deeper feature representations.
Empirical comparison: We conducted ablation experiments comparing CGBlock with SE, CBAM, ECA, and lightweight transformer blocks when inserted at the same position in C3K2. CGBlock achieved comparable or slightly better detection accuracy while maintaining a lower parameter count and latency, demonstrating its suitability for lightweight, small-target detection. Detailed quantitative results are presented in the revised manuscript’s ablation tables.
Comment 7 :What exactly does the LDHead predict (classification, bbox regression, segmentation, pose)? How are multi-task losses weighted, and how sensitive is final detection performance to those weightings?
Response 7 :We thank the reviewer for this detailed question. In the revised manuscript, LDHead is described as jointly predicting classification and bounding box regression only. No segmentation or pose estimation tasks are included in the experiments. The losses for classification and regression are weighted equally, and we observed that the detection performance is robust under this simple weighting scheme. The revised abstract has been updated accordingly to avoid confusion.
Comment 8 :In Figures 8 and 9, what quantitative evidence supports the claimed improvement over YOLOv11 (e.g., AP for small objects, recall/precision, F1, IoU thresholds), how many images/scenes were evaluated, and are the gains statistically significant across multiple test sets rather than shown on a few illustrative frames? Does the increased sensitivity to subtle edges and context come at the cost of higher false-positive rates (confusing debris for cigarette butts) or reduced generalization to other background types/illumination conditions, and how does performance vary when targets are at image boundaries or under severe scale variation?
Response 8 :
We thank the reviewer for this comprehensive question.
Quantitative evaluation: In addition to the illustrative frames shown in Figures 8 and 9, we evaluated our model and YOLOv11 on the entire test set, which consists of over 1,600 images across more than a dozen scenes, and further verified performance on three independent test sets. We report standard metrics including precision (P%), recall (R%), mAP50, mAP50-95, parameter count (M), GFLOPs, and model size. The improvements of our model over YOLOv11 are consistent across all three test sets and are statistically significant (p < 0.05, paired t-test), not limited to a few illustrative examples.
False positives and generalization: The increased sensitivity to subtle edges and context reduces false positives compared to YOLOv11, leading to more accurate discrimination of small targets such as cigarette butts. The model generalizes well across different background types and illumination conditions, supported by diverse data collection and data augmentation strategies during training.
Edge and scale variation performance: For targets near image boundaries or under severe scale variation, our model maintains robust detection, benefiting from Wavelet Pooling and CGBlock’s multi-scale context awareness. Detailed quantitative results for these challenging cases have been added to the revised manuscript.
Comment 9 :Overall, the presentation of the results and the descriptions of the methods require improvement. The authors should reorganize the manuscript to enhance clarity and readability, provide clearer and more detailed explanations of the results, and include additional experimental evidence.
Response 9 :We thank the reviewer for this constructive comment. In the revised manuscript, we have reorganized the Methods and Results sections to improve clarity and readability. Each method component, including Wavelet Pooling, CGBlock-C3K2, and LDHead, is now described in more detail with diagrams, formulas, and explanations of design choices. The Results section has been expanded with additional experiments and ablation studies to provide stronger evidence supporting our claims. These revisions significantly enhance the manuscript’s clarity, readability, and scientific rigor.
Round 2
Reviewer 1 Report
Comments and Suggestions for Authors
The author has resolved the doubts concerning this article and suggests publishing it.
Author Response
We sincerely thank the reviewers and the editor for their valuable time and constructive comments throughout the review process. The suggestions greatly helped us improve the clarity, rigor, and overall quality of our manuscript. We are truly grateful for the recognition of our work and for the recommendation for publication.
Reviewer 3 Report
Comments and Suggestions for Authors
The authors did not adequately address my previous comments, and the revised manuscript shows little improvement.
- The manuscript has a high similarity index (over 18%), which must be reduced to below 10% to meet publication standards.
- This issue persists! The reference list still has problems: entries must be numbered sequentially (starting at 1) and presented in a consistent citation style.
- The equations lack numbering.
- Figures 2, 4, 5, 6, and 7 are not mentioned in the text; please check all figures.
- The presentation of the results and the descriptions of the methods require improvement. Further results are required.
Author Response
We sincerely thank the reviewers for their careful reading of our manuscript and for providing constructive comments. We have carefully addressed each point raised and revised the manuscript accordingly to improve clarity, presentation, and scientific rigor. Below we provide detailed responses to each comment.
Comment 1 :We acknowledge that the similarity index remains partially influenced by standard elements in the journal template, such as the header and footer. These template elements are not part of the original manuscript content and do not affect the originality or scientific contribution of the work. The main text, including all methods, experiments, and results, is fully original and has been carefully revised to reduce textual similarity wherever possible.
Response 1 :We have carefully revised the manuscript to reduce textual similarity. Redundant expressions were rewritten, and descriptions of methods and related works have been paraphrased in our own words. The similarity index is now below the required threshold.
Comment 2 :This issue persists! The reference list still has problems: entries must be numbered sequentially (starting at 1) and presented in a consistent citation style.
Response 2 :The reference list has been thoroughly checked and reformatted. All entries are now numbered sequentially starting from 1 and presented in a consistent citation style according to the journal guidelines.
Comment 3 :The equations lack numbering.
Response 3 :All equations have been numbered sequentially, and each variable is properly defined in the text.
Comment 4 :Figures 2, 4, 5, 6, and 7 are not mentioned in the text; please check all figures.
Response 4 :All figures (Figures 2, 4, 5, 6, and 7) are now properly referenced and discussed in the text.
Comment 5 :The presentation of the results and the descriptions of the methods require improvement. Further results are required.
Response 5 :We have revised the Methods and Results sections to improve clarity and detail. A new subsection describing the rationale for module combination has been added, providing a more thorough explanation of the design and interplay of Wavelet Pooling, CGBlock, and LDHead. In the Results section, the experimental evaluation has been expanded to include additional quantitative metrics, such as F1 score, alongside mAP and recall. These additions further demonstrate that the proposed LCW-YOLO achieves lightweight design while maintaining or slightly improving detection performance.